# Dissecting surveying behavior of reactive microglia under chronic neurodegeneration

**Sunitha Subhramanian, Olga Bocharova, Natallia Makarava, Tarek Safadi, Ilia V Baskakov\***

Department of Neurobiology, University of Maryland School of Medicine, Baltimore, United States

## eLife Assessment

This **fundamental** study provides new evidence of a change in how microglia survey neurons during the chronic phase of neurodegeneration, which researchers studying neuroinflammation and its role in neurodegenerative disease should find interesting. In this research, using time-lapse imaging of acute brain slices from prion-affected mice, the researchers show that, unlike in healthy brains, microglia become reactive, lose their territorial boundaries, and become highly mobile, exhibiting "kiss-and-ride" behavior, migrating into brain tissue and forming reversible, transient body-to-body contact with neurons. The evidence is **compelling**, with well-executed time-lapse imaging, good quantitative analysis across several disease stages, pharmacological validation of P2Y6 involvement, and the very surprising finding that this mobile behavior persists after microglia are removed from the brain.

**\*For correspondence:**
Baskakov@som.umaryland.edu

**Abstract** In the healthy brain, microglia maintain homeostasis by continuously surveying neuronal health through highly dynamic processes that form purinergic junctions with neuronal somas. These mechanisms are finely tuned for the rapid detection of acute injuries. However, during the transition to a chronically reactive state in neurodegenerative diseases, microglial ramification decreases even as the need for neuronal monitoring escalates. How reactive microglia adapt their surveillance strategies under these conditions remains poorly understood. Using time-lapse imaging of acute brain slices from prion-infected mice, we identified a previously unrecognized mode of neuronal surveillance employed by reactive microglia. Unlike homeostatic microglia, which exhibit low somatic mobility and high process motility, enabling broad, simultaneous monitoring, reactive microglia display high somatic mobility. These cells actively migrate through the brain parenchyma, pausing to form direct and extensive body-to-body contacts with individual neurons. Contact durations ranged from minutes to several hours, often involving partial or full somatic envelopment, with transitions between these states being both frequent and reversible. Notably, reactive microglia exhibited sustained intracellular calcium bursts correlated with their increased mobility. Pharmacological inhibition of the P2Y6 receptor partially reduced microglial migration without disrupting their ability to form neuronal contacts. Furthermore, this highly mobile behavior persisted in acutely isolated reactive microglia in vitro, even in the absence of external stimuli, indicating that dynamic mobility is an intrinsic feature of the reactive phenotype. These findings reveal a fundamental shift in microglial surveillance architecture during chronic neurodegeneration – transforming from static, multi-neuron monitoring to dynamic, neuron-by-neuron engagement. This work uncovers a novel, adaptive strategy of microglial behavior with critical implications for understanding microglia–neuron interaction under chronic neurodegeneration.

## Introduction

Under homeostatic conditions, microglia continuously monitor neuronal health through specialized structures known as somatic purinergic junctions – sites where microglial processes make direct contact with neuronal cell bodies (*Cserép et al., 2020*). A key molecular component of these junctions is the P2Y12 receptor, which plays a critical role in mediating microglia–neuron communication (*Cserép et al., 2020*; *Cserép et al., 2022*). Homeostatic, ramified microglia exhibit highly dynamic behavior, extending and retracting their processes in a phenomenon termed *microglial motility* (*Cserép et al., 2022*; *Nebeling et al., 2023*; *Rotterman and Alvarez, 2020*; *Damani et al., 2011*). These transient process-to-soma contacts constitute the primary mode of neuronal surveillance in the healthy brain.

Under chronic neurodegenerative conditions, microglia acquire a sustained reactive phenotype (*Makarava et al., 2020a*; *Butovsky and Weiner, 2018*; *Krasemann et al., 2017*; *Keren-Shaul et al., 2017*). Reactive microglia differ markedly in morphology from their homeostatic counterparts, displaying an amoeboid shape with reduced process complexity. Both microglial ramification and P2Y12 receptor expression are significantly diminished as microglia transition to the reactive state (*Butovsky and Weiner, 2018*; *Krasemann et al., 2017*; *Zrzavy et al., 2017*; *Kenkhuis et al., 2022*; *Holtman et al., 2015*; *Maeda et al., 2021*). Paradoxically, this occurs at a time when the demand for neuronal surveillance intensifies due to increased neuronal stress and dysfunction. How microglia maintain effective surveillance of neurons under conditions of chronic neuroinflammation remains poorly understood.

Prion diseases, or transmissible spongiform encephalopathies, are fatal and infectious neurodegenerative disorders that affect both humans and animals (*Prusiner, 1998*). The hallmark pathological event in these diseases is the accumulation and spread of the misfolded, β-sheet-rich isoform of the prion protein (PrP$^{Sc}$) throughout the central nervous system. This process involves the templated conversion of the host's normal cellular prion protein (PrP$^{C}$) into the disease-associated PrP$^{Sc}$ conformation (*Legname et al., 2004*; *Makarava et al., 2018*). Chronic neuroinflammation is a prominent neuropathological feature of prion diseases, as well as other neurodegenerative diseases such as Alzheimer's and Parkinson's diseases (*Makarava et al., 2020a*; *Butovsky and Weiner, 2018*; *Krasemann et al., 2017*; *Keren-Shaul et al., 2017*; *Makarava et al., 2023*; *Makarava et al., 2019*). Notably, unlike animal models of other neurodegenerative diseases, prion-infected animals including mice develop authentic ultimately lethal neurodegenerative disease that recapitulates key features of the human prion disease (*Watts and Prusiner, 2014*; *Jeffrey et al., 2014*).

Previously, we demonstrated that in prion-infected mice, reactive microglia engage in a distinct form of microglia–neuron interaction characterized by partial envelopment of neuronal cell bodies (*Makarava et al., 2024*). These extensive body-to-body contacts differ from typical process-based surveillance. The enveloped neurons appear structurally intact, lack classical apoptotic markers, but exhibit functional impairments. This phenomenon is consistently observed across multiple brain regions exhibiting neuroinflammation in both prion-infected mice and human cases of sporadic Creutzfeldt–Jakob disease (*Makarava et al., 2024*). Strikingly, genetic ablation of P2Y12 increases the frequency of neuronal envelopment and accelerates disease progression in prion-infected mice (*Makarava et al., 2025*). In non-infected mice, P2Y12 deletion also increases the prevalence of microglia–neuron body-to-body interactions at the expense of process-to-body contacts (*Makarava et al., 2025*). These findings suggest that neuronal envelopment represents an alternative, P2Y12-independent mode of microglial surveillance, employed by reactive microglia under pathological conditions.

To date, quantification of neuronal envelopment has relied on fixed brain tissue (*Makarava et al., 2024*; *Makarava et al., 2025*; *Sinha et al., 2021*), leaving key questions about the temporal dynamics of these interactions unresolved. Although the frequency of envelopment increases with disease progression, it remains unclear whether individual interactions are sustained over time or whether reactive microglia sequentially engage multiple neurons through transient contacts.

The current study employed ex vivo time-lapse imaging of acute organotypic brain slices prepared from prion-infected mice to investigate the surveillance behavior of reactive microglia under chronic neurodegenerative conditions. Unlike homeostatic microglia, reactive myeloid cells lose territorial confinement. In contrast to homeostatic microglia, which exhibit low somatic mobility but high process motility enabling simultaneous monitoring of multiple neurons, reactive microglia display high somatic mobility. Reactive microglia move dynamically through tissue, engaging in sequential body-to-body contacts with individual neurons. This shift in behavior represents a fundamental reorganization of

microglial surveillance strategies in the diseased brain. Surprisingly, acutely isolated reactive microglia retained high-mobility in vitro in the absence of external stimuli, indicating that this dynamic behavior is an intrinsic feature of the reactive phenotype. The current studies reveal that in the reactive state, microglia adopt fundamentally different surveillance strategies to respond to the altered demands of the chronically inflamed and degenerating brain.

## Results

### Reactive microglia envelop neurons

To investigate microglial dynamics, we performed ex vivo time-lapse imaging using acute organo-typic brain slices prepared from Cx3cr1/EGFP mice (fractalkine receptor knockout/EGFP knock-in). These mice express enhanced green fluorescent protein (EGFP) under the control of the endogenous *Cx3cr1* promoter, restricting expression to myeloid cells. Previous studies have demonstrated that *Cx3cr1* deficiency does not influence disease pathogenesis, survival, or microglial activation in mice infected with the 22L or RML prion strains (*Striebel et al., 2016*). In agreement with these findings, we observed that *Cx3cr1* deficiency and EGFP knock-in in Cx3cr1/EGFP mice did not alter the incuba-tion time to disease following infection with mouse-adapted SSLOW or 22L prion strains, compared to wild-type (WT; C57BL/6J) controls (*Figure 1—figure supplement 1*). Moreover, PrP$^{Sc}$ accumula-tion levels were comparable between SSLOW-infected Cx3cr1/EGFP and WT mice (*Figure 1—figure supplement 1*).

For subsequent experiments, we selected the SSLOW strain due to its strong induction of neuroin-flammation and the shortest incubation time among mouse-adapted prion strains (*Makarava et al., 2020b*; *Makarava et al., 2021*). In prior studies, neuronal envelopment by microglia was consis-tently observed across all brain regions exhibiting prion-induced neuroinflammation (*Makarava et al., 2024*). In the present study, we focused on the cortex, where the temporal dynamics of neuronal envelopment during disease progression have been well characterized (*Makarava et al., 2024*). Exam-ination of fixed brain slices from SSLOW-infected Cx3cr1/EGFP mice at the clinical stage revealed extensive envelopment of neurons by Iba1$^+$ reactive microglia, characterized by prominent body-to-body contacts (*Figure 1a*), consistent with our previous report in WT mice (*Makarava et al., 2024*). Notably, the cortices of SSLOW-infected Cx3cr1/EGFP mice exhibited a high prevalence of neuronal envelopment events (*Figure 1—figure supplement 1*). In contrast, brain slices from age-matched, non-infected Cx3cr1/EGFP mice displayed microglia with a highly ramified morphology, primarily engaging in process-to-cell body interactions with neurons (*Figure 1a*, *Figure 1—figure supplement 1*), typical of homeostatic microglia.

### Reactive myeloid cells exhibit high mobility

To investigate microglial dynamics in prion-affected brains, ex vivo time-lapse imaging was performed on cerebral cortices of Cx3cr1/EGFP mice infected with the SSLOW prion strain via intraperitoneal (ip) injection. Imaging was conducted at three stages of disease progression: late sub-clinical (111–113 days post-inoculation, dpi), early clinical (125–128 dpi), and advanced (162–169 dpi). In mice inoculated with SSLOW via ip route, clinical onset typically occurs around 122 dpi (*Makarava et al., 2024*). Western blotting confirmed the presence of PrP$^{Sc}$ in the brains of mice examined at all three stages, with progressive accumulation of PrP$^{Sc}$ correlating with disease advancement (*Figure 2—figure supplement 1*). Age-matched non-infected Cx3cr1/EGFP mice (160–164 days old) served as healthy controls.

To be consistent with previously established terminology (*Smolders et al., 2019*), the term 'motility' will be used to refer to the movement of microglia processes, whereas the movement of cell bodies to new positions will be referred to by the term 'mobility'. Initial time-lapse recordings of acute brain slices from non-infected adult Cx3cr1/EGFP mice, acquired at a rate of one frame every 5 min, demonstrated that microglial morphology remained ramified – indicative of a homeostatic state – for at least 6 hr of continuous imaging (*Figure 2—video 1*). During this period, microglia largely remained within defined territories, displaying limited somatic mobility but sustained dynamic process activity, confirming cellular viability (*Figure 2—video 1*). However, noticeable photobleaching occurred after approximately 3 hr of imaging, with accelerated photobleaching observed at higher

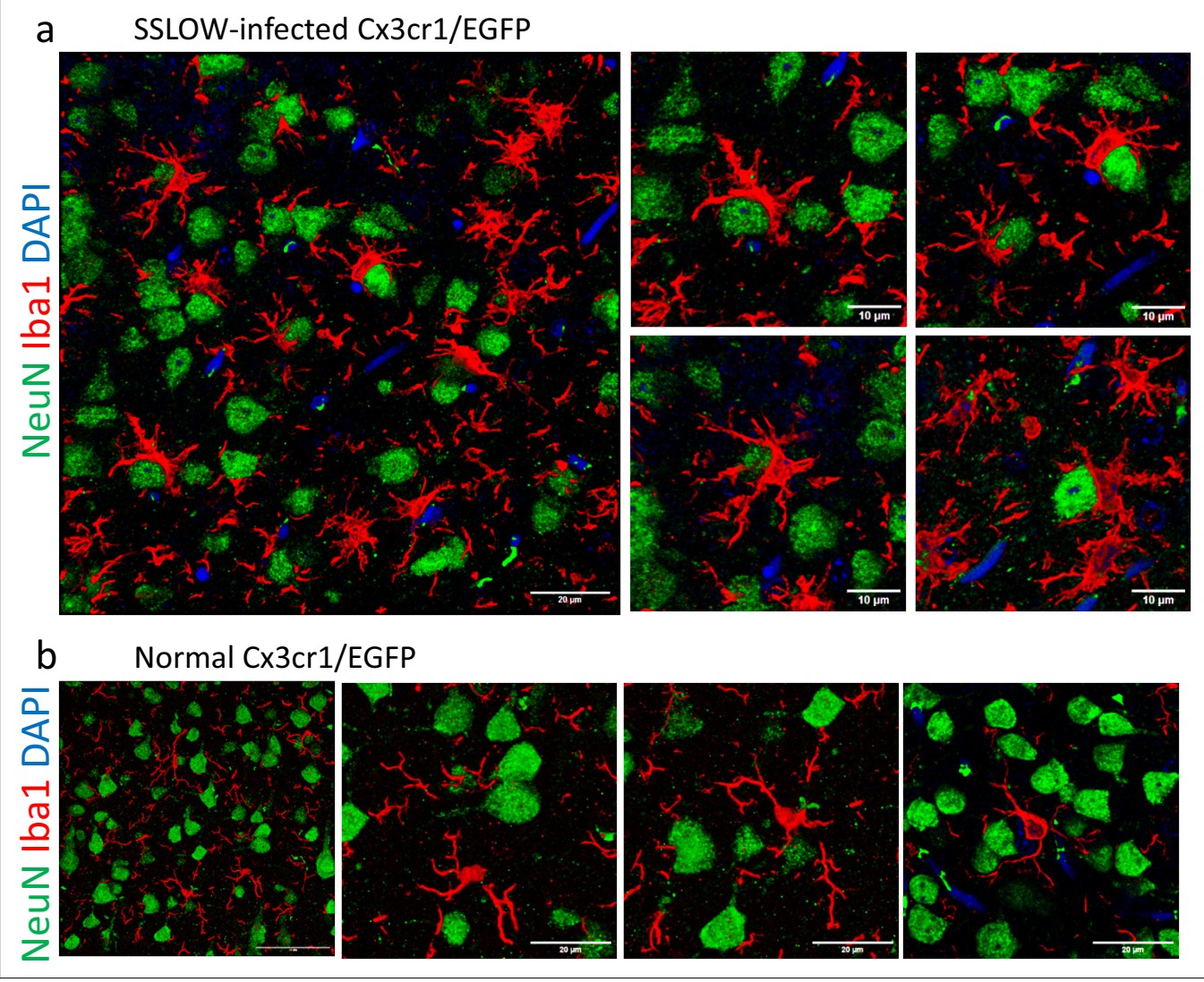

**Figure 1.** Iba1⁺ cells envelop neurons in SSLOW-infected brains. (**a**) Immunostaining for microglia (IBA1, red) and neurons (NeuN, green) showing neuronal envelopment by myeloid cells in cerebral cortex of Cx3cr1/EGFP mice infected by SSLOW via intraperitoneal route at the terminal stage of the disease. (**b**) Immunostaining for microglia (IBA1, red) and neurons (NeuN, green) of non-infected, age-matched Cx3cr1/EGFP mice. Confocal microscopy imaging followed by 3D reconstruction was used for both a and b.

The online version of this article includes the following source data and figure supplement(s) for figure 1:

**Figure supplement 1.** Prion pathogenesis is not changed in Cx3cr1/EGFP mice.

**Figure supplement 1—source data 1.** Western blot image of brains of WT and Cx3cr1/EGFP mice infected with SSLOW and age-matched controls.

**Figure supplement 1—source data 2.** Uncropped Western blot image of brains of WT and Cx3cr1/EGFP mice infected with SSLOW and age-matched controls.

imaging frequencies. Accordingly, all subsequent time-lapse videos were captured at a frame rate of one frame every 5 min.

Recent studies have shown that transcriptomic changes in slice cultures are most pronounced 1 day post-preparation (*Delbridge et al., 2020*), though morphological, functional, and gene expression alterations can begin as early as 4 hr post-slicing (*Ferrucci et al., 2024*). In contrast to the stable process motility observed in homeostatic microglia, EGFP⁺ cells in slices from SSLOW-infected animals exhibited both high process activity and active somatic translocation (*Figure 2—video 2*). To define the optimal imaging window for capturing dynamic behavior, we tracked individual cell movements across six consecutive 1-hr intervals following slice preparation. Microglial activity progressively

declined after 3–4 hr in both SSLOW-infected and control slices (*Figure 2—figure supplement 1*), prompting us to restrict all subsequent imaging experiments to a 3-hr window.

At all examined disease stages, EGFP+ cells in prion-infected brains displayed significantly increased mobility compared to controls, as reflected by elevated mean speed and total soma displacement (*Figure 2a–c*). A comprehensive heatmap analysis of mobility parameters, including soma displacement, speed, and directionality, demonstrated a progressive enhancement of microglial activity with advancing disease (*Figure 2d*).

## 'Kiss-and-ride' surveying behavior of reactive myeloid cells

Analysis of SSLOW-infected brain slices based on the distance traveled by EGFP+ cell somas revealed two distinct behavioral patterns: high mobility and low mobility. Low-mobility cells remained confined to a limited area, displaying minimal displacement and exhibiting localized, jiggling movements (*Figure 3a*). In contrast, high-mobility cells traversed significantly longer distances across the tissue (*Figure 3a, c, d*). These two phenotypes also differed in their mean directional change rate, a metric quantifying the average shift in movement direction over time (*Figure 3e*). Low-mobility cells exhibited higher directional change rates, consistent with their confined, wobbling behavior, whereas high-mobility cells demonstrated more linear and directed trajectories. To assess whether these patterns reflected genuine biological differences rather than inter-animal variation, we re-plotted the data as Superplots, in which mobility parameters for individual cells were averaged per animal. These analyses demonstrate that mobility metrics are highly consistent across animals within each group, indicating limited inter-animal variability (*Figure 3—figure supplement 1*).

Detailed tracking of high-mobility cells uncovered a distinctive, intermittent surveying pattern. EGFP+ cells extended processes in multiple directions, often simultaneously reaching out to several neurons. Subsequently, the cell somas translocated along some of these processes while others retracted (*Figure 4a, d*; *Figure 4—videos 1–4*). While migrating through the extracellular matrix, these cells paused to establish transient, soma-to-soma contact with neurons before resuming movement to engage with subsequent neurons. We refer to this dynamic pattern as 'kiss-and-ride' behavior. The duration of these transient contacts ranged from several minutes to over an hour (*Figure 4—videos 1–3*). Notably, this behavior was consistently observed across all three stages of disease progression: sub-clinical, early clinical, and advanced (*Figure 4a, d*; *Figure 4—videos 1–4*).

During a 3-hr imaging window, some EGFP+ cells were observed to contact up to six different neurons (*Figure 4c*; *Figure 4—video 2; Figure 4—video 5*), while others engaged with only one or two neurons, maintaining prolonged interactions with each (*Figure 4c, d*; *Figure 4—videos 1, 3, and 6*). Occasionally, a single EGFP+ cell simultaneously contacted or partially enveloped two or three neuronal somas (*Figure 4a*, *Figure 4—video 6*, *Figure 5a, b*; *Figure 5—video 1*). Interestingly, some neurons were sequentially surveyed by two different EGFP+ cells within the same 3-hr period (*Figure 4b*; *Figure 4—video 7*). Collectively, the behaviors of high-mobility EGFP+ cells underscore a lack of territorial restriction, contrasting with the spatially constrained dynamics typically seen in ramified microglia under homeostatic conditions.

## Enveloping behavior of reactive myeloid cells

Some low-mobility EGFP+ cells showed no apparent contact with neurons, whereas others maintained continuous contact with a single neuronal soma throughout the entire duration of time-lapse imaging (*Figure 4—video 6*, *Figure 5a*, *Figure 5—video 2*). Occasionally, low-mobility EGFP+ cells were observed slowly transitioning between neuronal somas or oscillating between two or three somas (*Figure 5a, b*; *Figure 5—video 1*, *Figure 4—video 1; Figure 4—video 6*). Body-to-body contact between EGFP+ myeloid cells and neurons was often sustained for extended periods and could progress to full envelopment of the neuronal soma (*Figures 4c and 5c*; *Figure 5—videos 3–6*). Notably, full envelopment did not necessarily culminate in phagocytosis; EGFP+ cells frequently reversed the process, returning from full envelopment to partial envelopment or contact (*Figure 4c*; *Figure 5—video 2*).

Previous analyses of fixed brain sections have shown that reactive myeloid cells selectively envelop neuronal cells, avoiding other cell types such as astrocytes or oligodendrocytes (*Makarava et al., 2024*; *Makarava et al., 2025*). In the current study, we attempted to visualize neuronal somas during time-lapse imaging using the calcium-sensitive dye Calbryte-590, following protocols from earlier

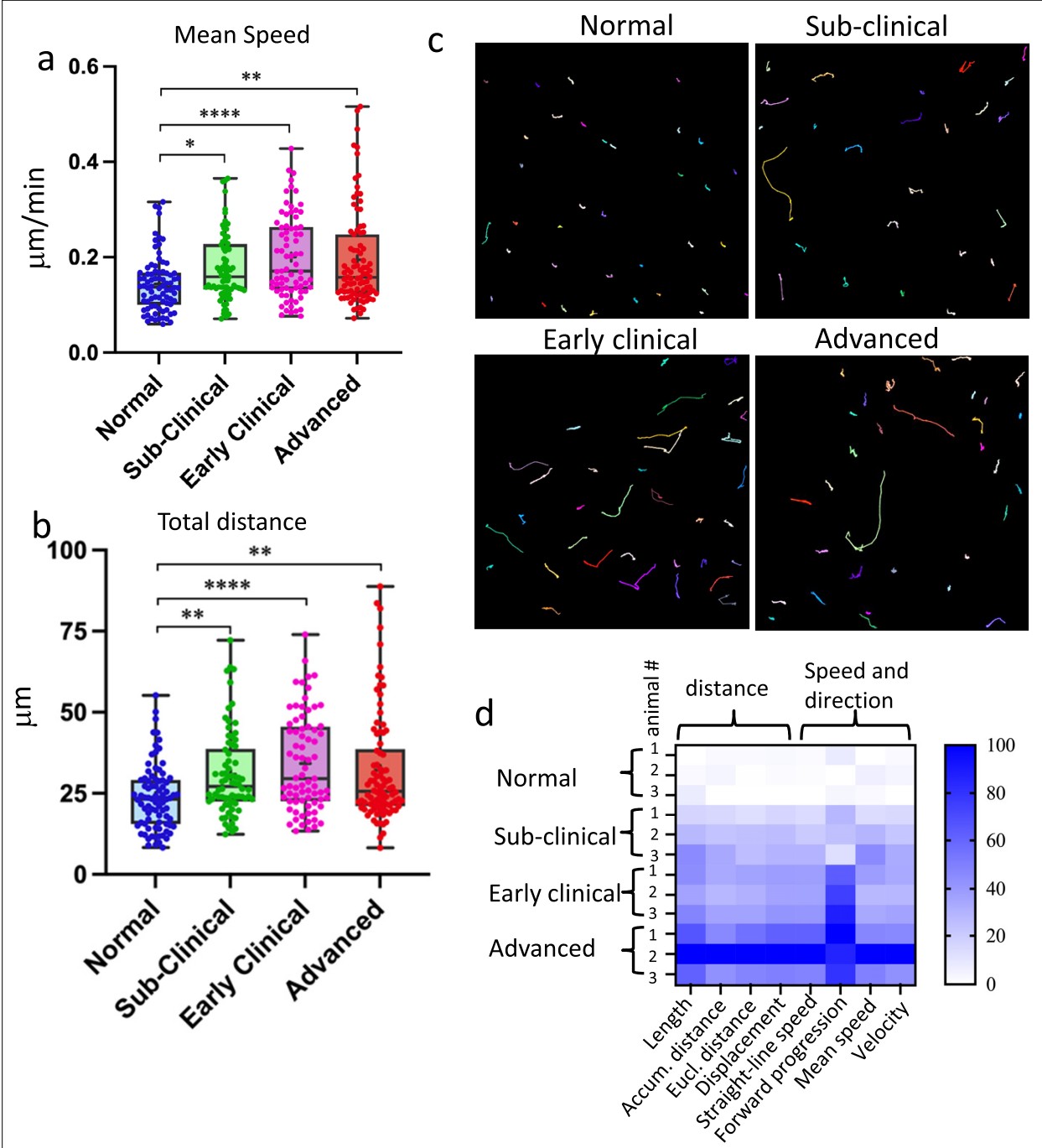

**Figure 2.** Reactive myeloid cells are highly mobile. Acute cerebral cortical slices were prepared using non-infected Cx3cr1/EGFP (normal) mice, or Cx3cr1/EGFP mice infected with SSLOW via ip route and examined at sub-clinical, early clinical, or advanced stages of the disease. Analysis of mean speed (**a**) and total distance (**b**) traveled by individual EGFP⁺ cells over a 3-hr period. The midline of the box-and-whisker plot denotes the median, the + represents the mean, and the ends of the box plot denote the 25th and 75th percentiles. (**c**) Examples of tracks recorded from EGFP⁺ cells. Colored lines represent tracks of individual cells recorded within a 3-hr period. (**d**) Principal component analysis of mobility parameters. $N = 3$ animals per group; $n = 70–90$ cells per group, *$p < 0.05$, **$p < 0.01$, ****$p < 0.0001$, ns – non-significant by non-parametric Kruskal–Wallis test with Dunn's multiple comparison.

The online version of this article includes the following video, source data, and figure supplement(s) for figure 2:

**Figure supplement 1.** Accumulation of PrPˢᶜ in SSLOW-infected Cx3cr1/EGFP mice.

**Figure supplement 1—source data 1.** Western blot image of brain homogenates from prion-infected Cx3cr1/EGFP mice used for time-lapse imaging experiments and to non-infected controls.

**Figure supplement 1—source data 2.** Uncropped Western blot image of brain homogenates from prion-infected Cx3cr1/EGFP mice used for time-

*Figure 2 continued on next page*

*Figure 2 continued*

lapse imaging experiments and to non-infected controls.

**Figure supplement 2.** Mobility of EGFP⁺ cells across six consecutive 1-hr intervals post-slicing.

**Figure 2—video 1.** Time-lapse video of acute brain slices from a normal, non-infected Cx3cr1/EGFP animal.

https://elifesciences.org/articles/107650/figures#fig2video1

**Figure 2—video 2.** Time-lapse imaging of acute brain slices from a SSLOW-infected Cx3cr1/EGFP mouse at the sub-clinical stage of the disease.

https://elifesciences.org/articles/107650/figures#fig2video2

research (*Cserép et al., 2022*). However, this proved challenging. Unlike neurons in non-infected animals, neurons in prion-infected slices exhibited markedly reduced Calbryte-590 fluorescence, suggesting dysfunction in calcium signaling. Despite the weak signal, some neuronal somas undergoing envelopment still showed detectable Calbryte-590 fluorescence in SSLOW-infected slices (*Figure 5—video 7*). Interestingly, brighter calcium puncta were frequently observed in association with reactive myeloid cells (*Figure 5—video 7*).

The same behavioral patterns observed in SSLOW-infected brains were also evident in brain slices from Cx3cr1/EGFP mice infected with the 22L mouse-adapted prion strain. These included both high-mobility EGFP⁺ cells exhibiting 'kiss-and-ride' interactions and low-mobility cells engaged in neuronal envelopment (*Figure 5—video 8*).

The interaction patterns between reactive myeloid cells and neurons that are seen in time-lapse imaging in dynamics could be captured at a much higher resolution using confocal microscopy imaging of fixed brain slices. Confocal imaging confirmed extensive body-to-body contact between Iba1⁺ cells and neurons, ranging from partial to full envelopment (*Figure 5d, e*). Some Iba1⁺ cells enveloped one neuron while simultaneously extending processes to contact or even envelop additional neurons. Cases were also observed where two Iba1⁺ cells partially enveloped a single neuron simultaneously. These patterns emphasize the profound body-to-body interactions between reactive myeloid cells and neurons in prion-infected brains, contrasting sharply with the predominantly process-mediated interactions seen under homeostatic conditions.

All behavioral patterns – 'kiss-and-ride', partial envelopment, and full envelopment – were consistently observed across all three stages of disease progression: preclinical, early clinical, and advanced. Notably, the percentage of high-mobility myeloid cells increased from the preclinical to early clinical stages (*Figure 3b*), possibly reflecting heightened surveillance demands in response to emerging neuronal dysfunction. At advanced stages, however, this percentage declined (*Figure 3b*), which may indicate a shift toward prolonged interactions with severely impaired neurons and/or the onset of intrinsic dysfunction within the reactive myeloid population.

## Morphology of both high- and low-mobility myeloid cells is consistent with reactive phenotype

To examine the morphological features associated with different patterns of EGFP⁺ cell behavior, morphological parameters were quantified for individual EGFP⁺ cells in each time frame, and then averaged across all time frames across the entire 3-hr imaging period to obtain a single mean value per cell. This statistical averaging approach accounts for the dynamic and continuously changing shapes of individual cells over time. Notable morphological differences were observed between cells from healthy animals and those from SSLOW-infected animals. Based on parameters such as cell radius, area, and perimeter, EGFP⁺ cells from SSLOW-infected animals exhibited a hypertrophic, amoeboid morphology – consistent with a reactive phenotype (*Figure 6a–c*). These morphological changes were evident in both high- and low-mobility cell populations, suggesting that regardless of mobility behavior, the cells adopted a reactive phenotype. A similar pattern was observed for the shape index, a dimensionless metric that quantifies overall cell shape (*Figure 6d*).

Interestingly, in comparison to low-mobility cells, the high-mobility cells from SSLOW-infected animals appeared to be closer to the cells from healthy controls (*Figure 6d*). This may reflect the elongated morphology adopted by mobile cells as they navigate the extracellular matrix, influencing the overall statistical representation of their shape. A trend toward lower values in cell radius, area, perimeter, and shape index with disease progression aligned with confocal imaging data, which illustrated

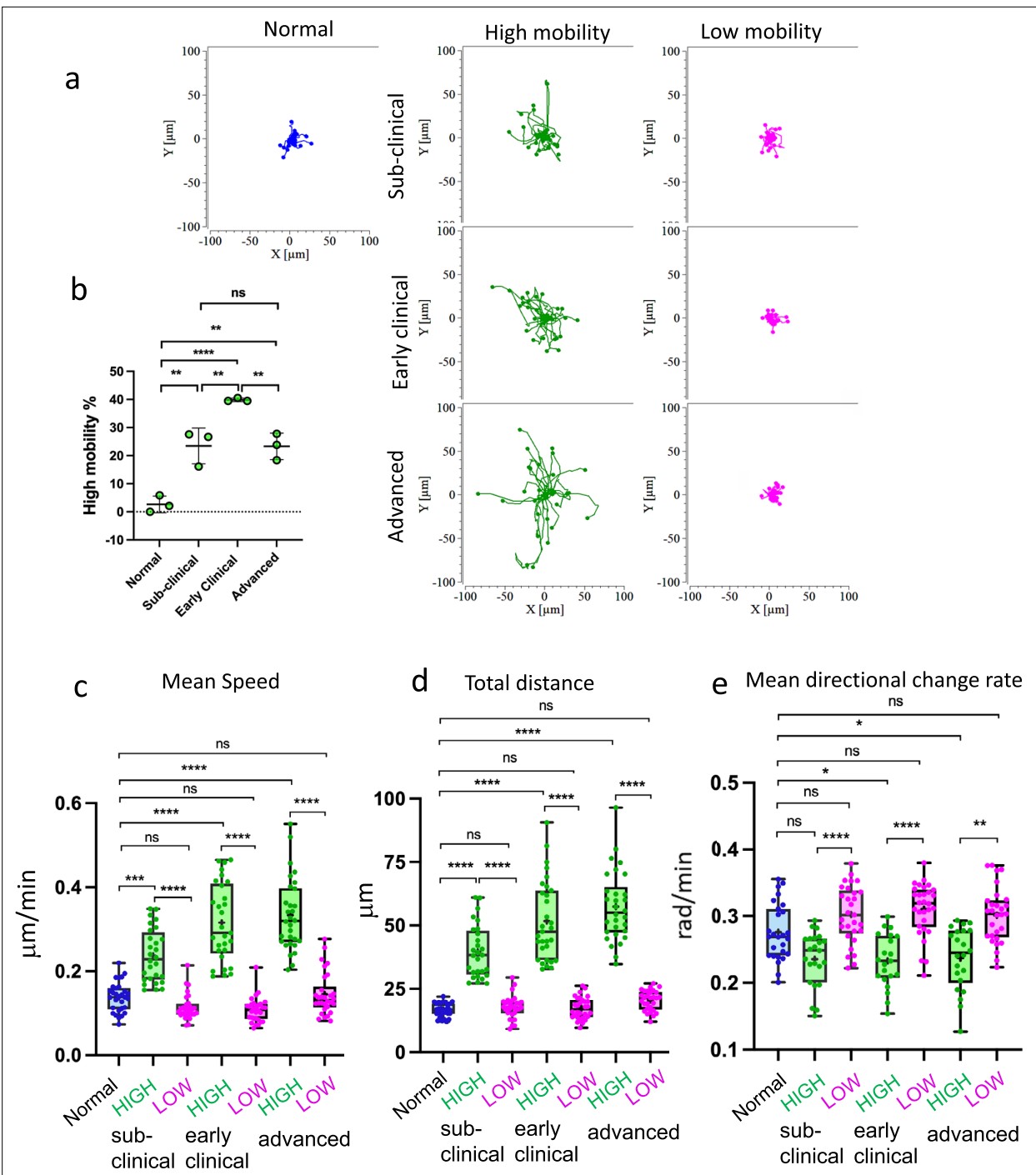

**Figure 3.** High- and low-mobility behavioral patterns of myeloid cells in prion-infected brains. Acute cerebral cortical slices were prepared using normal, non-infected Cx3cr1/EGFP mice, or SSLOW-infected Cx3cr1/EGFP mice and examined at the sub-clinical, early clinical, or advanced stages of the disease. (**a**) Rose plot of individual cell trajectories. In brain slices from SSLOW-infected animals, EGFP⁺ cells showed two behavioral patterns – with high and low mobility. (**b**) Change in the percentage of high-mobility EGFP⁺ cells with the disease progression. $N = 3$ animals per group. The data presented as means ± SD, **$p < 0.01$, ****$p < 0.0001$, ns – non-significant by Tukey's multiple comparisons test. Analysis of mean speed (**c**), total distance traveled over 3-hr period (**d**), and mean directional change rate (**e**) for individual high- and low-mobility EGFP⁺ cells in slices from SSLOW-infected mice at three disease stages, and normal mice. The midline of the box-and-whisker plot denotes the median, the + represents the mean, and the ends of the box plot denote the 25th and 75th percentiles. $N = 3$ animals per group; $n = 25–30$ cells per group, *$p < 0.05$, **$p < 0.01$, ***$p < 0.01$, ****$p < 0.0001$, ns – non-significant by non-parametric Kruskal–Wallis test with Dunn's multiple comparison.

The online version of this article includes the following figure supplement(s) for figure 3:

**Figure supplement 1.** Inter-animal variation in high- and low-mobility behavioral phenotypes.

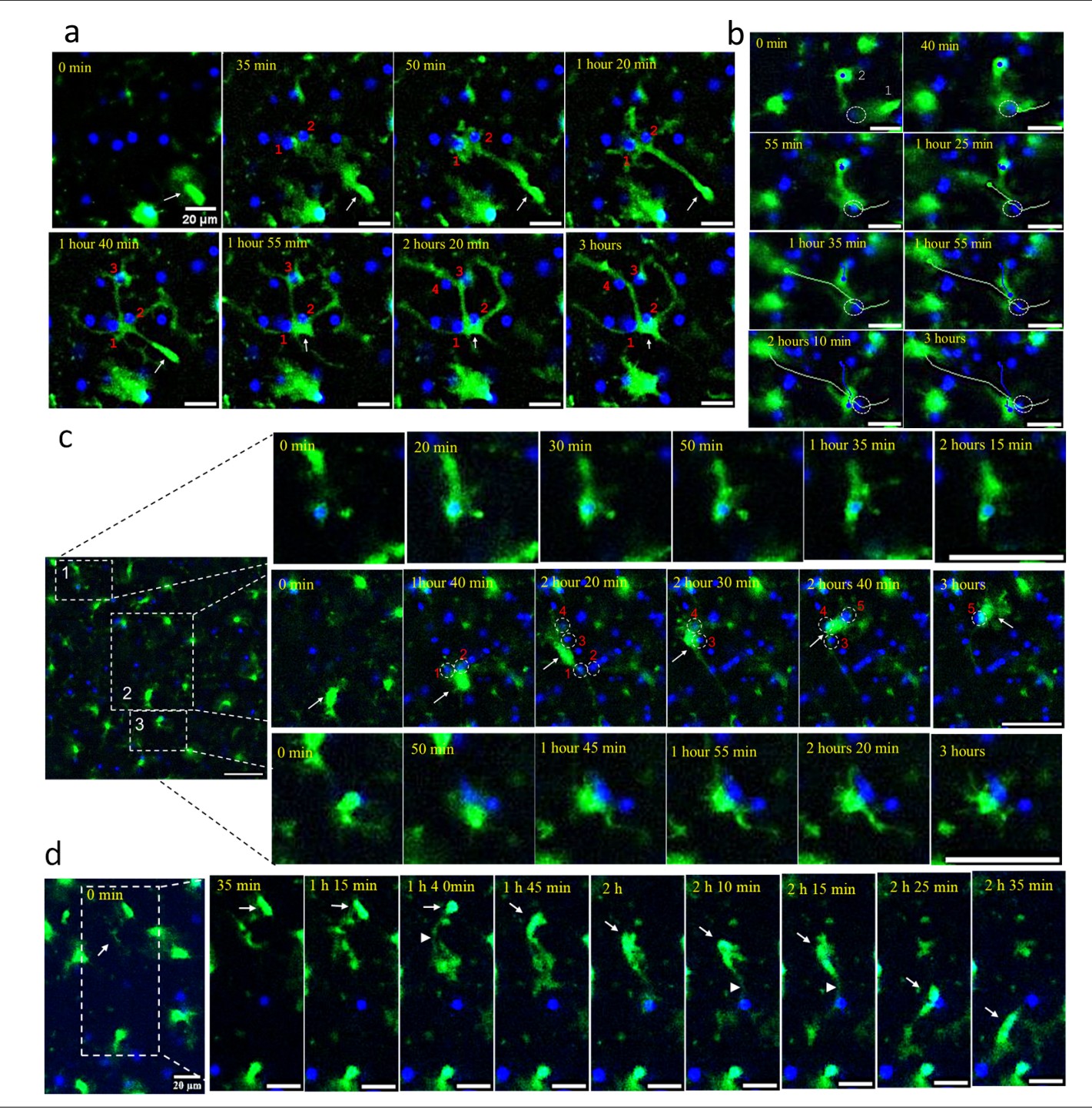

**Figure 4.** Behavioral patterns of high-mobility reactive myeloid cells. Time-lapse imaging of acute cerebral cortical slices of SSLOW-infected Cx3cr1/EGFP mice captured at the early clinical (**a**), advanced (**b, d**), and sub-clinical (**c**) stages of disease. (**a**) (*Figure 4—video 1*): An EGFP+ cell (indicated by an arrow) extends a process toward neurons #1 and #2, migrates along this process, and subsequently envelops both neurons while simultaneously extending processes toward neurons #3, #4, and #5. (**b**) (*Figure 4—video 7*): EGFP+ cell #1 migrates toward and surveys a neuron (circled), then departs. Subsequently, a second EGFP+ cell (#2) migrates to and interacts with the same neuron. White and blue curves trace the respective migration paths of the two cells. (**c**) Upper panels: An EGFP+ cell approaches, envelops, and then retracts from a neuron. Middle panels: A migrating EGFP+ cell (indicated by an arrow) surveys five distinct neurons (circled), interacting simultaneously with neurons #1 and #2, followed by #3 and #4. Lower panels: An EGFP+ cell maintains prolonged contact with a neuron. (**d**) (*Figure 4—video 3*): An EGFP+ cell (arrow) migrates across the field, initially extending processes (arrowhead) toward a neuron. The cell body then translocates along these processes, briefly contacts one neuron, and continues movement toward another. All videos were recorded over a 3-hr period with 5-min intervals. Nuclei were visualized using Hoechst staining. Scale bars = 20 μm.

*Figure 4 continued on next page*

*Figure 4 continued*

The online version of this article includes the following video(s) for figure 4:

**Figure 4—video 1.** Time-lapse imaging of acute brain slices from a SSLOW-infected Cx3cr1/EGFP mouse at the early clinical stage of the disease.
https://elifesciences.org/articles/107650/figures#fig4video1

**Figure 4—video 2.** Time-lapse imaging of acute brain slices from a SSLOW-infected Cx3cr1/EGFP mouse at the sub-clinical stage of the disease.
https://elifesciences.org/articles/107650/figures#fig4video2

**Figure 4—video 3.** Time-lapse imaging of acute brain slices from a SSLOW-infected Cx3cr1-EGFP mouse at the advanced stage of the disease.
https://elifesciences.org/articles/107650/figures#fig4video3

**Figure 4—video 4.** Time-lapse imaging of acute brain slices from a SSLOW-infected Cx3cr1/EGFP mouse at the advanced stage of the disease.
https://elifesciences.org/articles/107650/figures#fig4video4

**Figure 4—video 5.** Time-lapse imaging of acute brain slices from a SSLOW-infected Cx3cr1/EGFP mouse at the sub-clinical stage of the disease.
https://elifesciences.org/articles/107650/figures#fig4video5

**Figure 4—video 6.** Time-lapse imaging of acute brain slices from a SSLOW-infected Cx3cr1/EGFP mouse at the early clinical stage of the disease.
https://elifesciences.org/articles/107650/figures#fig4video6

**Figure 4—video 7.** Time-lapse imaging of acute brain slices from a SSLOW-infected Cx3cr1/EGFP mouse at the advanced stage of the disease.
https://elifesciences.org/articles/107650/figures#fig4video7

a more pronounced amoeboid morphology and reduced ramification at advanced disease stages (*Figures 5d, e and 6a–d*).

Overall, the statistical averaging of morphological parameters supports the classification of cells based on the clinical status of the animals from which brain slices were derived. Myeloid cells with similar mobility but different clinical backgrounds (e.g., normal vs. SSLOW low-mobility) exhibited clear morphological distinctions, whereas cells with differing mobility within the same pathological condition (SSLOW high- vs. low-mobility) were morphologically similar.

## High-mobility myeloid cells exhibit sustained Ca$^{2+}$ bursts

Calcium signaling plays a pivotal role in microglial activation and migration, particularly in response to brain injury or neuronal damage (*Eichhoff et al., 2011*; *Umpierre et al., 2020*; *Umpierre et al., 2024*). Previous studies have demonstrated that activated microglia exhibit sustained calcium bursts that correlate with their migratory behavior (*Du et al., 2022*). Using Calbryte-590, we observed sustained calcium bursts in EGFP$^+$ cells, especially in highly mobile populations, within brain slices from SSLOW-infected mice (*Figure 7a*).

Due to the time-lapse imaging interval of one frame every 5 min, we were unable to precisely quantify the duration or frequency of individual calcium bursts. Nonetheless, high-mobility cells consistently exhibited elevated calcium signals throughout the entire 3-hr imaging period (*Figure 7b, c*, *Figure 7—videos 1–3*). These signals were primarily localized to the soma, although occasional bursts were also observed within cellular processes (*Figure 7—videos 1–3*). EGFP$^+$ cells engaged in neuronal envelopment also showed calcium activity, albeit at lower intensities than highly mobile cells (*Figure 7b, c*, *Figure 7—videos 2 and 4*).

## Inhibition of P2Y6 receptor reduces the mobility of reactive myeloid cells

Calcium transients in microglia are regulated by the P2Y6 receptor (*Umpierre et al., 2024*). Activation of this receptor by its endogenous ligand, UDP, has been implicated in multiple microglial functions, including migration, phagocytosis of damaged neurons, and clearance of apoptotic debris and Aβ plaques (*Neher et al., 2014*, *Umpierre et al., 2024*; *Puigdellívol et al., 2021*; *Milde et al., 2021*; *Koizumi et al., 2007*). In addition to P2Y6, several members of P2Y receptors and ATP-gated P2X channels contribute to microglial surveillance, activation, motility, and phagocytic responses. We therefore examined the expression of P2ry6, P2ry13, P2*rx*7, and P2*rx*4 alongside activation markers Tlr2, Cd68, and Trem2. Bulk brain tissue analysis revealed that all examined genes were upregulated in SSLOW-infected mice relative to controls (*Figure 8—figure supplement 1*). However, because microglial proliferation markedly increases cell numbers during disease progression (*Makarava et al., 2024*; *Gómez-Nicola et al., 2013*; *Gómez-Nicola et al., 2014*; *Gomez-Nicola and Perry, 2015*),

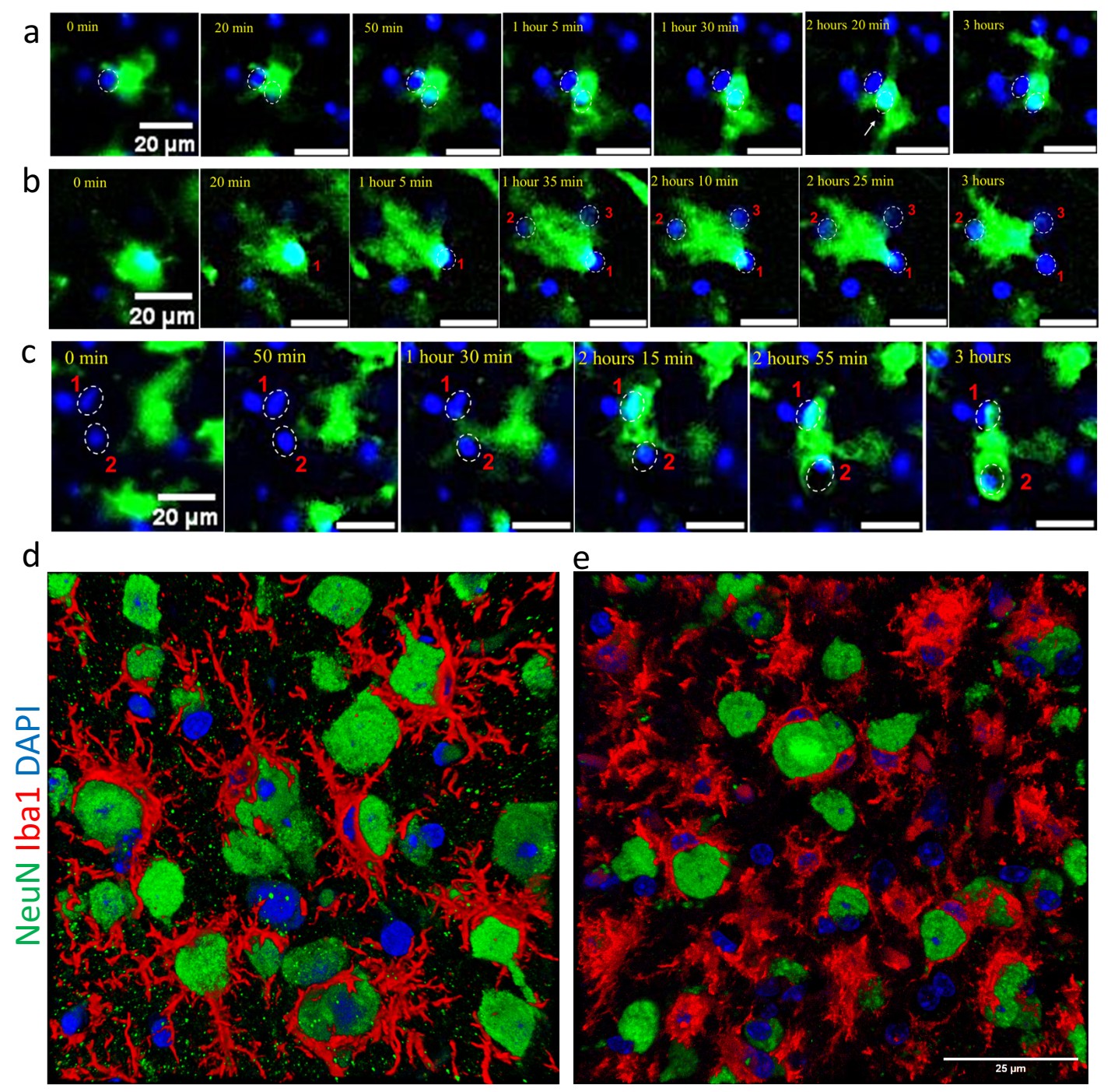

**Figure 5.** Behavioral patterns of low-mobility reactive myeloid cells. Time-lapse imaging of acute cerebral cortical slices of SSLOW-infected Cx3cr1/EGFP mice, captured at early clinical (**a, b**) and advanced (**c**) stages of disease. (**a**) An EGFP⁺ cell exhibits prolonged interactions simultaneously with two neurons. (**b**) An EGFP⁺ cell shows sustained envelopment of neuron #1, then initiates simultaneous envelopment of neuron #2 while maintaining contact with neuron #1 and possibly neuron #3. (**c**) An EGFP⁺ cell moves toward neurons #1 and #2, partially envelops neuron #1, and fully envelops neuron #2. All videos were recorded over a 3-hr period with 5-min intervals. Nuclei were visualized using Hoechst staining. Scale bars = 20 μm. (**d, e**) 3D reconstructions from confocal microscopy images of fixed cerebral cortical slices immunostained for myeloid cells (IBA1, red) and neurons (NeuN, green) in WT mice infected with SSLOW via the intraperitoneal route, analyzed at early clinical and advanced disease stages. Scale bar = 25 μm.

The online version of this article includes the following video(s) for figure 5:

**Figure 5—video 1.** 3D reconstruction of time-lapse imaging of acute brain slices from a SSLOW-infected Cx3cr1/EGFP mouse at the advanced stage of the disease.

*Figure 5 continued on next page*

*Figure 5 continued*

https://elifesciences.org/articles/107650/figures#fig5video1

**Figure 5—video 2.** Time-lapse imaging of acute brain slices from a SSLOW-infected Cx3cr1/EGFP mouse at the early clinical stage of the disease.
https://elifesciences.org/articles/107650/figures#fig5video2

**Figure 5—video 3.** Time-lapse imaging of acute brain slices from a SSLOW-infected *Cx3cr1/EGFP* mouse at the early clinical stage of the disease.
https://elifesciences.org/articles/107650/figures#fig5video3

**Figure 5—video 4.** Time-lapse imaging of acute brain slices from a SSLOW-infected Cx3cr1/EGFP mouse at the early clinical stage of the disease.
https://elifesciences.org/articles/107650/figures#fig5video4

**Figure 5—video 5.** Time-lapse imaging of acute brain slices from a SSLOW-infected Cx3cr1/EGFP mouse at the advanced stage of the disease.
https://elifesciences.org/articles/107650/figures#fig5video5

**Figure 5—video 6.** 3D reconstruction of time-lapse imaging of acute brain slices from a SSLOW-infected Cx3cr1/EGFP mouse at the advanced stage of the disease.
https://elifesciences.org/articles/107650/figures#fig5video6

**Figure 5—video 7.** Time-lapse imaging of acute brain slices from a SSLOW-infected Cx3cr1/EGFP mouse at the advanced stage of the disease.
https://elifesciences.org/articles/107650/figures#fig5video7

**Figure 5—video 8.** Time-lapse imaging of acute brain slices from a 22L-infected Cx3cr1/EGFP mouse at the early clinical stage of the disease.
https://elifesciences.org/articles/107650/figures#fig5video8

bulk expression changes may not accurately reflect per-cell expression levels. To account for this, we normalized gene expression to the microglia-specific marker Tmem119. Although Tmem119 is considered a marker of homeostatic microglia, its per-cell expression remains relatively stable during prion disease progression (*Makarava et al., 2025*). After normalization, *Tlr2*, *Cd68*, and *Trem2* were upregulated approximately 10-, 6-, and 4-fold, respectively, whereas P2 receptor gene expression showed more modest increases: *P2ry6* by 3-fold, *P2ry13* by 2-fold, and *P2rx7* by 1.3-fold, while P2*rx4* remained unchanged (*Figure 8—figure supplement 1*). Given its role in calcium signaling, the magnitude of upregulation and its specific expression in microglia, P2Y6 was selected for further functional analysis.

To assess the role of P2Y6 signaling in the dynamics of reactive myeloid cells, we acutely inhibited the receptor using the selective antagonist MRS-2578. This was performed on brain slices obtained at the early clinical disease stage, when the proportion of highly mobile EGFP⁺ cells was maximal (*Figure 3b*).

MRS-2578 treatment reduced both mean cell speed and total distance traveled by EGFP⁺ cells in SSLOW-infected slices compared to mock-treated controls (*Figure 8a*). However, mobility did not decrease to the levels observed in uninfected slices, suggesting a partial rather than complete effect of P2Y6 inhibition (*Figure 8a*). In MRS-2578-treated slices, the majority of reactive EGFP⁺ cells were observed either enveloping neuronal somata or remaining closely associated with them (*Figure 8—video 1; Figure 8—video 2*). In some cases, individual EGFP⁺ cells were seen alternating between neighboring neuronal bodies, sequentially enveloping each soma (*Figure 8b*; *Figure 8—video 2*). These findings suggest that P2Y6 signaling facilitates long-range migration of reactive myeloid cells. While its inhibition diminishes overall mobility, it does not abolish localized movements or neuron-associated behaviors such as soma-to-soma interactions.

## Reactive myeloid cells retain elevated basal mobility in vitro in the absence of external stimuli

In homeostatic microglia, the extension of cellular processes and migration toward injury sites are driven by neuronal activity and extracellular cues such as ATP, ADP, and UDP. It is generally presumed that similar cues govern the dynamics of chronically reactive microglia; however, this remains unverified. To directly test whether environmental stimuli are required for the mobility of reactive myeloid cells, we examined their behavior in vitro under stimulus-free conditions.

Myeloid cells were acutely isolated from SSLOW-infected and uninfected Cx3cr1/EGFP mice using CD11b-coated magnetic beads. Basal mobility was then assessed in vitro without any exogenous stimulation. Remarkably, cells from SSLOW-infected mice exhibited significantly higher basal mobility than those from control animals, as reflected by increased speed and displacement (*Figure 9a, b, d*).

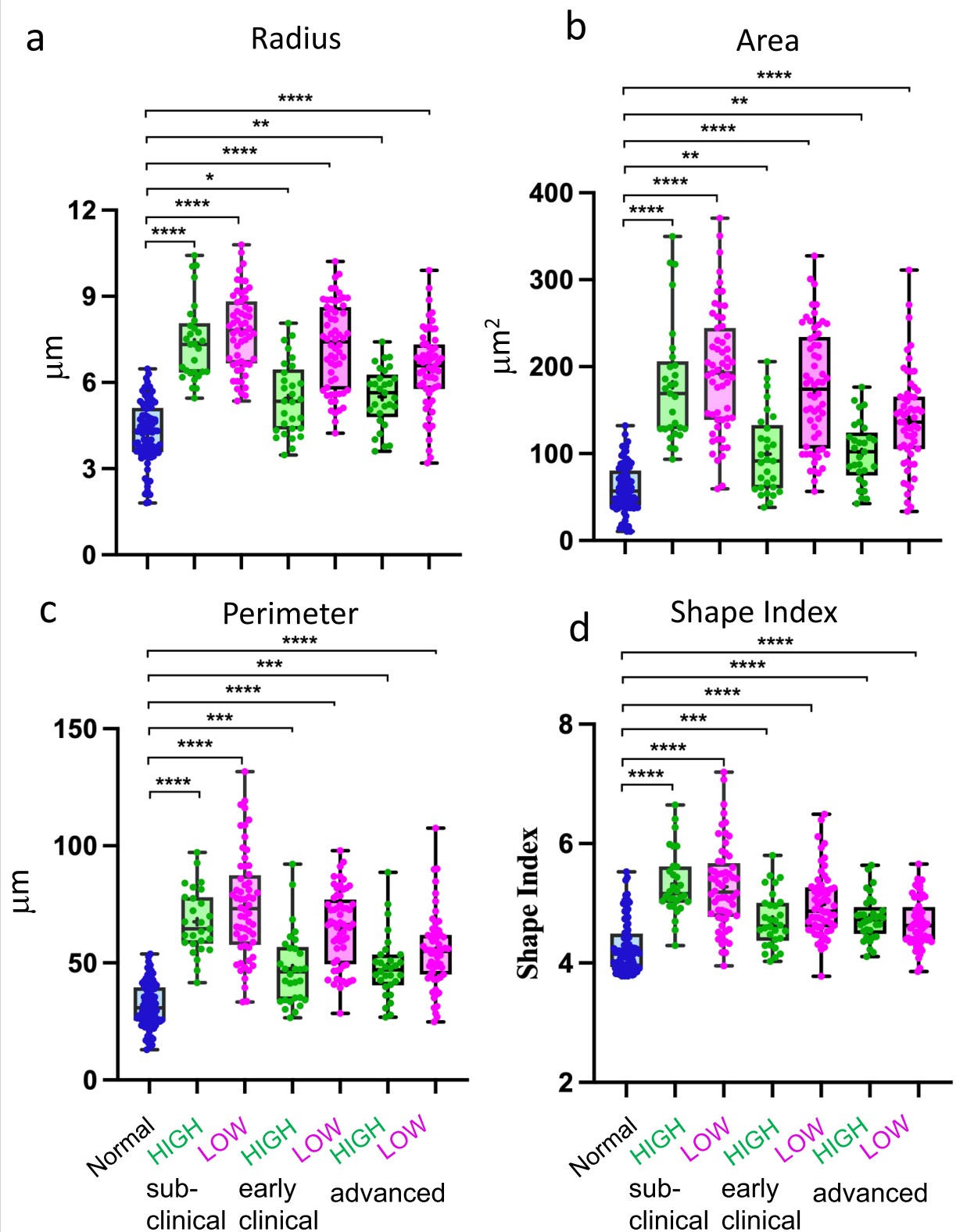

**Figure 6.** Analysis of cell morphology. Acute cerebral cortical slices were prepared from normal, non-infected Cx3cr1/EGFP mice and from SSLOW-infected Cx3cr1/EGFP mice at the sub-clinical, early clinical, or advanced stages of the disease. Analysis of cell radius (**a**), cell area (**b**), cell perimeter (**c**), and shape index (**d**) of high- and low-mobility EGFP+ cells in slices from SSLOW-infected mice at three disease stages and normal mice. The midline of the box-and-whisker plot denotes the median, the + represents the mean, and the ends of the box plot denote the 25th and 75th percentiles. *N* = 3

*Figure 6 continued on next page*

*Figure 6 continued*

animals per group; *n* = 30–90 cells per group, *p < 0.05, **p < 0.01, ***p < 0.01, ****p < 0.0001, ns – non-significant by non-parametric Kruskal–Wallis test with Dunn's multiple comparison.

Co-culture with N2a neuroblastoma cells did not alter the mobility of control myeloid cells. However, it modestly enhanced the mobility of reactive myeloid cells compared to monoculture conditions (*Figure 9a, b, d*). Notably, this increase was accompanied by a decrease in directional change rate, suggesting a shift toward more directed and less stochastic movement in the presence of N2a cells (*Figure 9c*). Together, these results indicate that reactive myeloid cells possess an intrinsically elevated basal mobility that does not require exogenous stimulation. Moreover, despite their heightened autonomous activity, these cells remain responsive to environmental cues that do not elicit similar behavioral changes in homeostatic microglia.

## Myeloid cells that envelop neurons are TMEM119⁺ and P2Y12⁺

In Cx3cr1/EGFP mice, EGFP is expressed in all myeloid cells, including both brain-resident and infiltrating populations. To determine the identity of cells involved in neuronal envelopment, brain sections from SSLOW-infected Cx3cr1/EGFP mice were immunostained for TMEM119 and P2Y12, markers characteristic of resident microglia. Although P2Y12 expression in individual microglia is known to be markedly downregulated as prion disease progresses (*Makarava et al., 2025*; *Slota et al., 2022*), IBA1⁺ cells exhibiting neuronal envelopment were found to be positive for both TMEM119 and P2Y12. These findings indicate that resident microglia are responsible for the envelopment behavior (*Figure 8—figure supplements 2 and 3*).

## Discussion

In the healthy brain, microglia continuously monitor neuronal activity via highly dynamic processes that extend and retract to form purinergic junctions with neuronal somas (*Cserép et al., 2020*; *Cserép et al., 2022*; *Nebeling et al., 2023*; *Berki et al., 2024*). These surveillance mechanisms in the homeostatic state are primarily tuned to detect acute injuries, as microglia respond to changes in neuronal activity and environmental signals such as ATP, ADP, and UDP released by damaged neurons (*Rotterman and Alvarez, 2020*; *Damani et al., 2011*; *Koizumi et al., 2007*; *Haynes et al., 2006*; *Maeda et al., 2010*; *Fekete et al., 2018*; *Ohsawa et al., 2007*; *Eyo et al., 2018*). In contrast to acute or focal injuries, neuronal dysfunction in chronic conditions evolves gradually and spreads across broader regions. It is commonly assumed that reactive microglia continue to rely on the same cues and surveillance strategies as in the homeostatic state. However, during the transition to a reactive phenotype in the context of chronic neurodegeneration, both the complexity and number of microglial processes involved in surveillance decrease, even as the demand for neuronal monitoring intensifies. Additionally, expression of P2Y12, a key receptor for sensing ATP and ADP at purinergic junctions, is significantly reduced (*Butovsky and Weiner, 2018*; *Krasemann et al., 2017*; *Zrzavy et al., 2017*; *Kenkhuis et al., 2022*; *Holtman et al., 2015*; *Maeda et al., 2021*). This raises a critical question: How do reactive microglia adapt their surveillance strategies to meet the increasing demands of chronic neurodegeneration?

Our findings suggest that, in the reactive state, microglia adopt a distinct approach to neuronal monitoring. In the homeostatic brain, individual microglial cells occupy well-defined territories, extending their processes to simultaneously contact multiple neurons. By contrast, in the reactive state, myeloid cells lose this territorial organization, becoming highly mobile and migrating through the extracellular matrix. They pause intermittently to form direct body-to-body contacts, typically with one neuron at a time. This movement begins with process extension, followed by somal translocation along the path of the extended process.

On average, each myeloid cell surveyed one neuron per hour. However, the duration of contact varied considerably – from a few minutes to several hours – as did the extent of neuronal envelopment. Partial envelopment of neuronal somas was the most common form of interaction, though full envelopment was also observed. Notably, full envelopment was often reversible, transitioning back to partial envelopment or complete retraction. These ongoing morphological changes – including the appearance, disappearance, and reappearance of intercellular contacts – underscore the highly

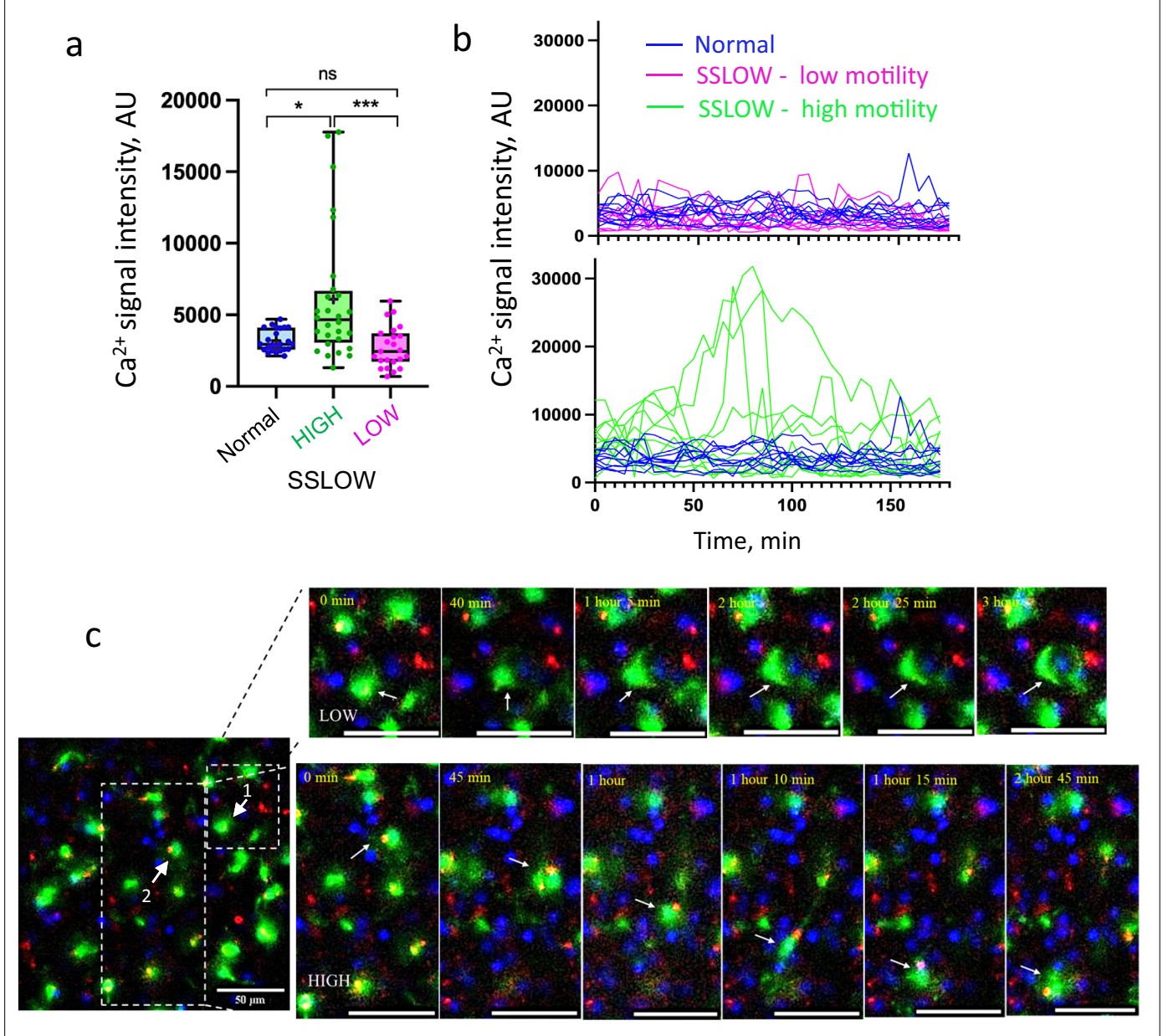

**Figure 7.** Sustained Ca²⁺ bursts in high-mobility myeloid cells. Acute cerebral cortical slices were prepared from non-infected Cx3cr1/EGFP mice and from SSLOW-infected Cx3cr1/EGFP mice at the early clinical stage of disease. (**a**) Quantification of signal intensity of Ca²⁺ puncta within high- or low-mobility EGFP⁺ cells. Data from normal, non-infected mice are shown for reference. Sustained Ca²⁺ bursts were detected using Calbryte-590 AM and averaged per cell over a 3-hr period. $N = 3$ animals per group; $n = 24$–28 cells per group. *p < 0.05, **p < 0.01, ns = not significant by non-parametric Kruskal–Wallis test with Dunn's multiple comparisons. (**b**) Changes in Ca²⁺ signal intensity in individual EGFP⁺ cells over the 3-hr imaging session. Images were acquired at 5-min intervals. (**c**) Time-lapse imaging of acute brain slice from a SSLOW-infected Cx3cr1/EGFP mouse recorded over 3-hr. Upper panels (*Figure 7—video 2*): EGFP⁺ cell (#1) envelops a neuronal soma and exhibits low Ca²⁺ activity. Lower panels: highly mobile EGFP⁺ cell (#2) displays sustained somatic Ca²⁺ bursts. Scale bars = 50 µm.

The online version of this article includes the following video(s) for figure 7:

**Figure 7—video 1.** Time-lapse imaging of acute brain slices from a SSLOW-infected Cx3cr1/EGFP mouse at the advanced stage of the disease.
https://elifesciences.org/articles/107650/figures#fig7video1

**Figure 7—video 2.** Time-lapse imaging of acute brain slices from a SSLOW-infected Cx3cr1/EGFP mouse at the advanced stage of the disease.
https://elifesciences.org/articles/107650/figures#fig7video2

**Figure 7—video 3.** Time-lapse imaging of acute brain slices from a SSLOW-infected Cx3cr1/EGFP mouse at the advanced stage of the disease.
https://elifesciences.org/articles/107650/figures#fig7video3

*Figure 7 continued on next page*

*Figure 7 continued*

**Figure 7—video 4.** Time-lapse imaging of acute brain slices from a SSLOW-infected Cx3cr1/EGFP mouse at the advanced stage of the disease.
https://elifesciences.org/articles/107650/figures#fig7video4

dynamic nature of communication between reactive myeloid cells and neuronal somas. Although individual contacts were fluid, they could be maintained for several hours. It is tempting to speculate that the duration and degree of envelopment may reflect both the extent of neuronal damage and the complexity of the underlying decision-making processes governing neuronal fate.

Reactive myeloid cells frequently interacted with two or three neuronal somas simultaneously, exhibiting oscillatory or 'wobbling' movements between them. This behavior suggests a capacity for multitarget surveillance. Moreover, within a 3-hr window, multiple myeloid cells could be observed sequentially interacting with the same neuron, indicating again a breakdown of territorial organization.

To navigate the extracellular matrix, an amoeboid morphology offers clear advantages over a ramified one. In the reactive state, microglia upregulate matrix-degrading enzymes that facilitate migration by clearing paths through the adhesive matrix (*Lively and Schlichter, 2013*). The mobility pattern of reactive myeloid cells observed in our study resembles that of embryonic mouse microglia, which alternate between phases of process extension and subsequent somal translocation along the axis of the extended processes (*Smolders et al., 2017*). This mechanical similarity suggests that reactive microglia may engage migratory mechanisms akin to those used during developmental stages to support their surveillance functions. However, unlike their embryonic counterparts, reactive myeloid

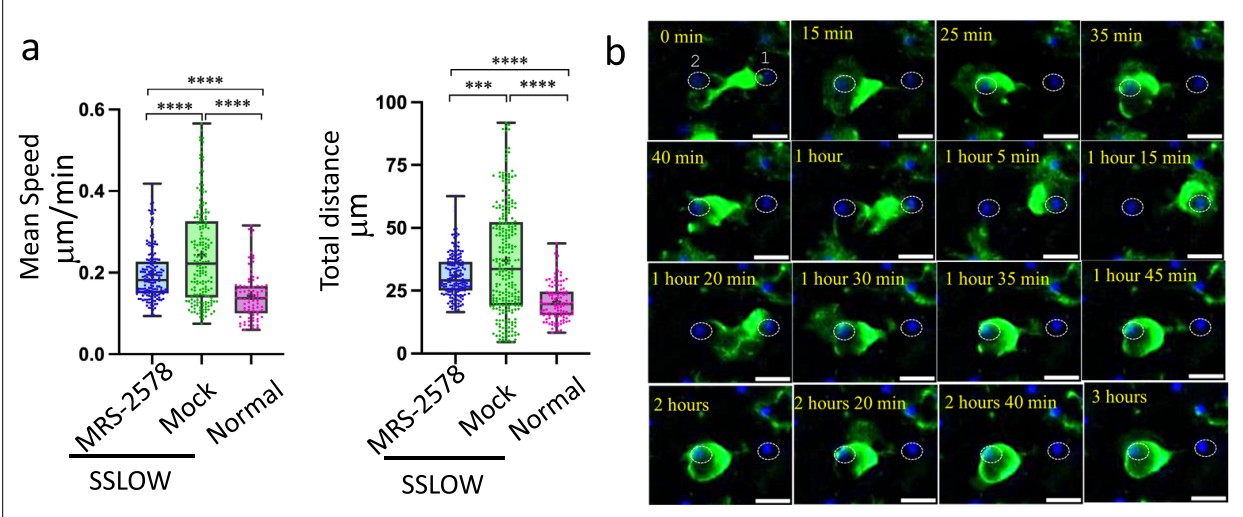

**Figure 8.** Inhibition of the P2Y6 receptor reduces motility of reactive myeloid cells. Acute cerebral cortical slices were prepared from non-infected Cx3cr1/EGFP mice and from SSLOW-infected Cx3cr1/EGFP mice at the early clinical stage of disease. (a) Quantification of mean speand total distance traveled by individual EGFP+ cells over a 3-hr period in brain slices from SSLOW-infected mice treated with either MRS-2578 (2 μM) or vehicle control. Data from normal, non-infected mice are shown for reference. N = 3 animals per group; n = 100–250 cells per group. **p < 0.001, ***p < 0.0001 by Tukey's multiple comparisons test. (b) Time-lapse imaging of MRS-2578-treated acute brain slice from a SSLOW-infected Cx3cr1/EGFP mouse recorded over 3 hr. The EGFP+ cell exhibits bidirectional movement between two neuronal somas (labeled #1 and #2), sequentially enveloping each soma. Scale bars = 20 μm.

The online version of this article includes the following video and figure supplement(s) for figure 8:

**Figure supplement 1.** Gene expression analysis in SSLOW-infected mice.

**Figure supplement 2.** Envelopment of neurons by P2Y12-positive microglia.

**Figure supplement 3.** Envelopment of neurons by TMEM119-positive microglia.

**Figure 8—video 1.** Time-lapse imaging of acute brain slices from a SSLOW-infected Cx3cr1/EGFP mouse at the early clinical stage of disease.
https://elifesciences.org/articles/107650/figures#fig8video1

**Figure 8—video 2.** Time-lapse imaging of acute brain slices from a SSLOW-infected Cx3cr1/EGFP mouse at an early clinical stage of disease.
https://elifesciences.org/articles/107650/figures#fig8video2

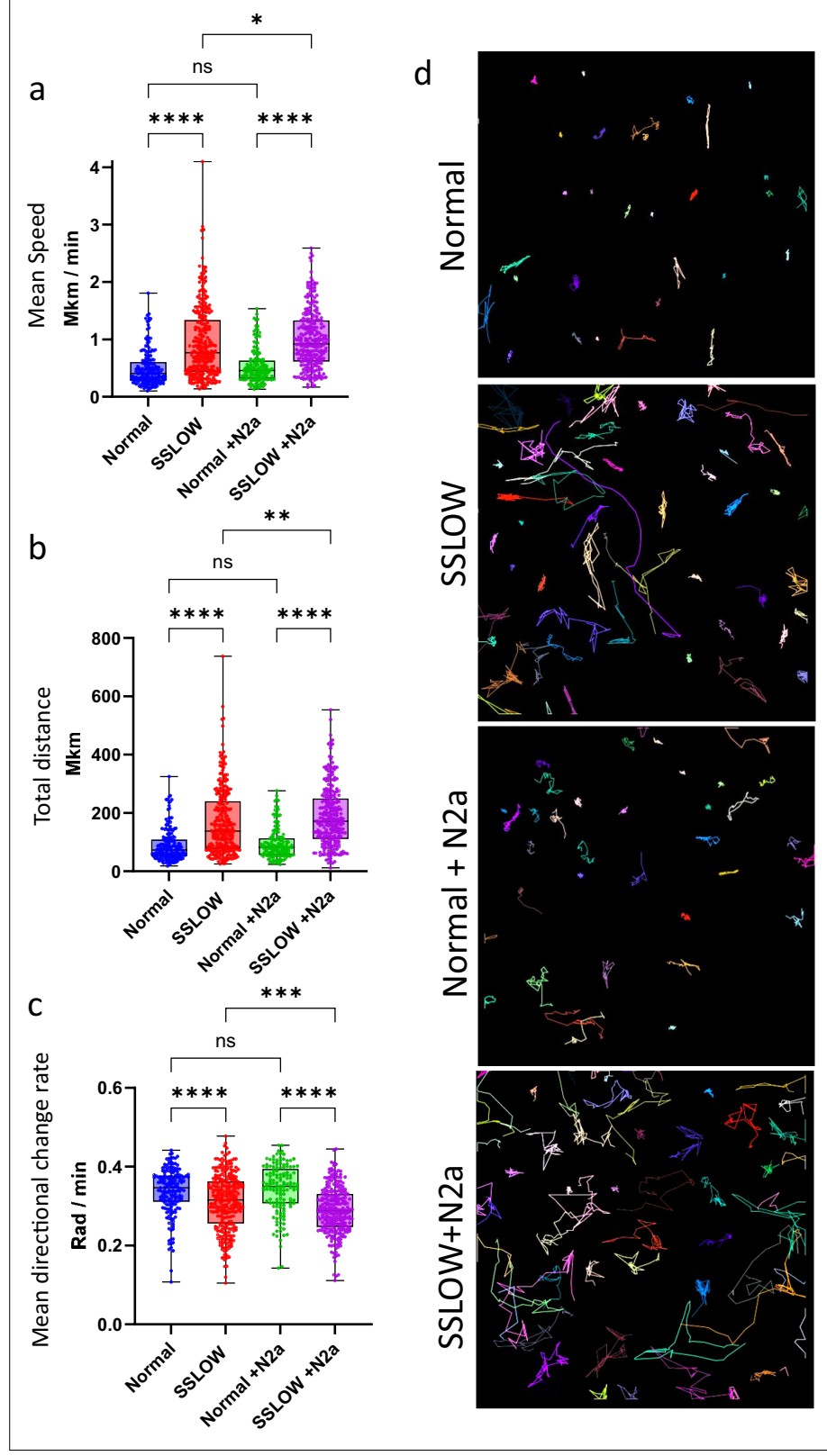

**Figure 9.** In vitro analysis of CD11b[+] cell mobility. CD11b[+] cells were acutely isolated from Cx3cr1/EGFP mice at the clinical stage of SSLOW infection or from non-infected Cx3cr1/EGFP mice. Quantification of cell mobility parameters, including mean speed (**a**), total track distance (**b**), and mean directional change rate (**c**) of CD11b[+]/EGFP[+] cells over a 3-hr period, in the presence or absence of N2a cells. In box-and-whisker plots, the midline

*Figure 9 continued on next page*

*Figure 9 continued*

indicates the median, the 'x' denotes the mean, and the box limits represent the 25th and 75th percentiles. *N* = 9 and 6 animals for non-infected and SSLOW-infected group, respectively. *n* = 178–354 cells per group. Statistical significance: p < 0.05; p < 0.01; *p < 0.001; **p < 0.0001; ns, not significant by non-parametric Kruskal–Wallis test with Dunn's multiple comparison. (**d**) Representative cell tracks of CD11b$^+$/EGFP$^+$ cells over 3 hr. Colored lines indicate trajectories of individual cells.

cells appear to actively scan their environment to select among multiple potential neuronal targets, indicating a more decision-oriented migratory behavior.

In our study, the average speed of the highly mobile microglial population ranged from 13.2 µm/hr at the sub-clinical stage to 19.1 µm/hr at the advanced disease stage. These velocities are lower than those reported for hippocampal microglia at postnatal day 2 (~36 µm/hr) and cortical microglia at embryonic day 17.5 (~25 µm/hr) (*Eyo et al., 2016*). It is important to note that our measurements likely underestimate actual migration speeds due to the low frequency of frame acquisition in time-lapse experiments and potential movements occurring between frames that were not captured.

In this work, we classified cells into two categories – high and low mobility – based on their behavior during the 3-hr observation period. It remains unclear whether these categories represent distinct phenotypic subpopulations. Given the limited time window, the observed mobility differences may reflect transient stages within the dynamic surveying behavior, rather than stable phenotypic traits. At the advanced disease stage, the proportion of highly mobile cells declined, potentially due to prolonged interactions between myeloid cells and neurons. This observation raises compelling questions for future investigation, including whether distinct phenotypes emerge among reactive myeloid cells as a consequence of neuronal interactions, and what the functional implications of such differences might be.

The activation and migration of homeostatic microglia in response to acute neuronal injury are known to be mediated by calcium signaling (*Eichhoff et al., 2011*; *Umpierre et al., 2020*; *Umpierre et al., 2024*; *Du et al., 2022*). In the present study, sustained calcium bursts were observed in association with the migratory activity of high-mobility myeloid cells, suggesting that calcium signaling is crucial for migration in the reactive state too. However, unlike the brief calcium transients typically observed in homeostatic microglia, which last up to a minute, reactive myeloid cells exhibited elevated calcium levels for extended durations of at least 3 hr.

In microglia, calcium signaling is regulated in part by the metabotropic P2Y6 receptor (*Umpierre et al., 2024*). Activation of P2Y6 by its endogenous ligand UDP has been implicated in various microglial functions, including migration, phagocytosis of damaged neurons, and clearance of apoptotic bodies and Aβ plaques (*Neher et al., 2014*, *Umpierre et al., 2024*; *Puigdellívol et al., 2021*; *Milde et al., 2021*; *Koizumi et al., 2007*). In the present study, we found that the P2Y6 antagonist MRS-2578 partially suppressed the migration of reactive myeloid cells but did not prevent their association with neurons. These findings suggest that while P2Y6 signaling facilitates the mobility of myeloid cells, it does not mediate their physical interaction with neurons. This points to a specific role for P2Y6 in the surveillance behavior of reactive myeloid cells under conditions of chronic neurodegeneration. Multiple purinergic receptors, including P2Y1, P2Y2, P2Y4, P2Y6, P2Y12, P2RY13, P2X4, and P2X7, have been implicated in microglial chemotaxis, phagocytosis, and directed movement in response to acute injury (*Smolders et al., 2019*; *Franco-Bocanegra et al., 2019*). In prion-infected mice, we observed modest upregulation of *P2ry6*, *P2ry13*, and *P2rx7*, with no detectable change in *P2rx4* expression, whereas prior studies have reported downregulation of P2Y12 receptor when normalized per cell (*Makarava et al., 2025*). Nevertheless, the observation that MRS-2578 only partially inhibited cell movement supports the idea that multiple, potentially redundant pathways govern microglial mobility.

A particularly intriguing finding of this study is that reactive myeloid cells maintained elevated mobility upon acute isolation in vitro, even in the absence of external stimuli or neuronal activity. This indicates that heightened mobility is an intrinsic property of the reactive phenotype, in stark contrast to homeostatic microglia, which typically require external cues to initiate soma migration. These results imply that signaling mechanisms beyond purinergic pathways contribute to the enhanced mobility of reactive myeloid cells. It is possible that this behavior is sustained through autocrine or paracrine signaling involving proinflammatory factors secreted by reactive microglia themselves. Supporting

this notion, recent work has demonstrated that IFNγ promotes microglial migration in the adult mouse cortex (*Boghozian et al., 2023*), and genes associated with IFNγ signaling are upregulated in microglia from prion-infected mice (*Nazmi et al., 2019*). Future studies will be needed to determine whether such autonomous signaling mechanisms underlie the highly mobile phenotype.

Quantitatively, the mean speed and total distance traveled by reactive myeloid cells in vitro were two- to threefold greater than those observed in acute brain slices. This is not unexpected, given that in situ, cells must navigate the extracellular matrix and often pause to establish direct contacts with neurons. Notably, co-culturing reactive microglia with N2a neuronal cells increased their mobility even further. In the presence of N2a cells, reactive microglia exhibited more directed and less stochastic movement patterns. These observations suggest that, in a reactive state, myeloid cells retain a heightened responsiveness to environmental cues that do not elicit similar behavioral changes in homeostatic microglia.

Several limitations of this study warrant discussion. In Cx3cr1/EGFP mice, all myeloid cells, including monocytes and macrophages, express EGFP, raising questions about the specific identity of the myeloid cells that establish extensive contacts with neurons. Our current and previous work demonstrated that myeloid cells enveloping neurons are positive for TMEM119 and P2Y12 (*Makarava et al., 2025*), markers generally associated with resident microglia. Additionally, prior studies have shown that microglial expansion in prion diseases primarily results from the proliferation of resident microglia, with minimal contribution from peripheral myeloid cell recruitment (*Gómez-Nicola et al., 2013*; *Gómez-Nicola et al., 2014*). Nevertheless, infiltration of peripheral myeloid cells and their potential role in neuronal surveillance cannot be completely ruled out.

It is unclear whether reactive myeloid cells in other neurodegenerative diseases employ similar strategies for neuronal surveillance. Furthermore, the molecular mechanisms and docking pathways mediating the formation of close body-to-body contacts between reactive myeloid cells and neurons are not yet understood. In earlier work, we found that knockout of CD11b, a component of complement receptor 3 involved in the phagocytosis of newborn neurons during neurodevelopment, does not affect the prevalence of neuronal envelopment or the progression of prion diseases (*Makarava et al., 2024*). Conversely, knockout of the P2Y12 receptor, which mediates purinergic junctions between microglial processes and neuronal soma, led to an increased prevalence of neuronal envelopment (*Makarava et al., 2025*). These findings suggest that P2Y12 is not required for body-to-body contact formation; rather, its absence facilitates neuronal envelopment by microglia.

The role of the high-mobility phenotype of reactive microglia in neuronal health and disease progression is poorly understood too. In a previous study, the onset of neuronal envelopment followed a decline in cellular levels of Grin1, a subunit of the NMDA receptor essential for synaptic plasticity. Reactive microglia were observed to envelop Grin1-deficient neurons, suggesting that myeloid cells respond to neuronal dysfunction (*Makarava et al., 2024*). Notably, P2Y12 knockout increased the prevalence of neuronal envelopment and accelerated disease progression (*Makarava et al., 2025*). Collectively, these observations suggest that while microglial envelopment may represent an adaptive response to heightened neuronal surveillance demands, excessive envelopment, as seen in the absence of P2Y12, appears to be maladaptive.

Cx3cr1 is a chemokine receptor expressed by microglia that binds its neuronal ligand Cx3cl1, which exists in both membrane-bound and soluble forms (*Bazan et al., 1997*). The Cx3cr1–Cx3cl1 axis plays a role in homeostatic functions such as immune surveillance, chemotaxis, and phagocytosis (*Raoul et al., 2008*). While CX3CR1 is known to regulate microglial process dynamics and migration, whether Cx3cr1 signaling reduces or accelerates microglial motility remains a matter of debate (*Liang et al., 2009*; *Wagner et al., 2024*). In the present study, we observed robust neuronal envelopment by reactive microglia in Cx3cr1/EGFP mice, which lack functional Cx3cr1, indicating that microglia-neuronal docking does not require Cx3cr1-Cx3cl1 signaling. Furthermore, microglia isolated from prion-infected Cx3cr1/EGFP and C57BL/6J mice exhibited comparably high-mobility in vitro, suggesting that Cx3cr1 contributes minimally to the regulation of surveillance behavior reactive microglia.

To summarize, the transformation of myeloid cells into a reactive phenotype involves a fundamental reorganization of microglial surveillance strategies. In the homeostatic state, microglia monitor a limited territory surrounding their location and require chemical cues to initiate migration toward sites of injury. In contrast, reactive myeloid cells exhibit high intrinsic mobility and actively search for neurons. These cells adopt a distinct surveillance pattern characterized by migration from one neuron

to another, pausing to form extensive, body-to-body contacts – typically with a single neuron at a time. Prion-infected animals develop a lethal form of authentic prion disease. The observation of neuronal envelopment by reactive myeloid cells in both individuals with sporadic CJD and prion-infected mice underscores that this aspect of human disease pathology is faithfully recapitulated in prion-infected mice. To our knowledge, this study represents the first report of microglial surveillance behavior in the context of a bona fide neurodegenerative disease rather than a disease model.

## Materials and methods

### Animals

B6.129P2(Cg)-*Cx3cr1*<sup>tm1Litt</sup>/J mice (Strain 005582, The Jackson Laboratory) were purchased from the Jackson laboratory and bred in house. B6.129P2(Cg)-*Cx3cr1*<sup>tm1Litt</sup>/J have an EGFP sequence replacing the first 390 bp of the coding exon of the chemokine (C-X3-C motif) receptor 1 (*Cx3cr1*) gene.

Brain-derived material for inoculations was prepared from terminally ill SSLOW- or 22L-infected wild type (WT, C57BL/6J) mice as 10% (wt/vol) brain homogenate (BH) in PBS, pH 7.4, using glass/ Teflon homogenizers attached to a cordless 12 V compact drill (*Makarava et al., 2012*). Immediately before inoculation, the inoculum was further dispersed by 30 s indirect sonication at ~200 W in a microplate horn of a sonicator (Qsonica, Newtown, CT) and diluted to 1% in PBS, pH 7.4. Mice were randomly assigned to prion-infected or non-infected control groups. B6.129P2(Cg)-*Cx3cr1*<sup>tm1Litt</sup>/J and WT mice were inoculated with 200 µl 1% BH intraperitoneally under 3% isoflurane anesthesia. Alternatively, mice were inoculated with 20 µl 1% BH intracranially (Figure S1a). In accordance with IACUC regulations, the inoculation status of animals was clearly indicated on each cage to ensure compliance with approved protocols. To minimize subjectivity, all procedures involving animal handling, including inoculation and clinical scoring, were performed by dedicated personnel who were not involved in the experimental design. This separation of duties helped reduce the potential for conscious or unconscious bias. Animals were scored weekly using four categories, each of which was graded using score '0–3', with '3' being the most severe impairment. The scoring categories were: clasping hind legs, posture (kyphosis, rigid tail, rearing difficulties), mobility (difficulties in ambulation and navigating the cage edge), and gait (keeping balance while walking, wobbly gait, disorientation, lethargy). The animals were deemed symptomatic when they displayed a consistent increase in the combined score starting from '4'. The mice were considered terminal when they were unable to rear and/or lost 20% of their weight. The following animal groups were used for main experiments: SSLOW-inoculated B6.129P2(Cg)-*Cx3cr1*<sup>tm1Litt</sup>/J mice at late sub-clinical (111–113 dpi, dpi), early clinical (125–128 dpi), and advanced stages (162–169 dpi), and non-infected 160- to 164-day-old B6.129P2(Cg)-*Cx3cr1*<sup>tm-1Litt</sup>/J mice. Animals of both sexes in random ratios were used in all experiments (*Table 1*).

### Antibodies

Primary antibodies used for immunofluorescence, immunohistochemistry, and immunoblotting were as follows: rabbit polyclonal anti-IBA1 (#013–27691, FUJIFILM Wako Chemicals USA, Richmond, VA); goat polyclonal anti-IBA1 (#NB100-1028, Novus, Centennial, CO); mouse monoclonal anti-NeuN, clone A60 (#MAB377, Millipore-Sigma, Burlington, MA); mouse monoclonal anti-prion protein, clone SAF-84 (#189775, Cayman, Ann Arbor, MI); rabbit monoclonal anti-prion protein, clone 3D17 (#ZRB1268, Millipore-Sigma); rabbit polyclonal anti-P2Y12 (#55043A, Anaspec, Fremont, CA); rabbit monoclonal anti-TMEM119, clone E3E10 (#90840, Cell Signaling, Danvers, MA). The secondary

**Table 1.** Sex of animals used in experiments.

| Figure # | Normal | Sub-clinical | Early clinical | Advanced |
|---|---|---|---|---|
| 2, 3, 6 | 2M/1F | 3M | 1M/2F | 2M/1F |
| 7 | 3F | | 2M/1F | |
| | | Early clinical: MRS-2578 | | Mock |
| 8 | 2M/1F | 1M/2F | | 2M/1F |
| 9 | 3M/6F | | 1M/5F | |

antibodies for immunofluorescence were Alexa Fluor 488-, 546-, and 647-labeled (Thermo Fisher Scientific, Waltham, MA).

## Acute brain slice preparation and ex vivo time-lapse imaging

Before euthanizing an animal, sterile PTFE hydrophilic membrane inserts (PICM0RG50, Sigma) of 0.4 µm pore size were kept in a 6-well plate supplemented with serum-free culture media and incubated at 37°C and 5% $CO_2$ for 1–2 hr in a standard cell culture incubator. For acute slice preparation, the whole mouse brain was removed from the skull and was immersed in ice-cold oxygenated ACSF media (LRE-S-LSG10001, Ecocyte Bioscience LLC) saturated with 95% $O_2$ and 5% $CO_2$. The cerebellum and olfactory bulb were cut off, and the remaining portion of the brain was glued to the bottom of the specimen holder within the tissue slicing chamber of the Vibratome (Leica VT1200) filled with oxygenated ACSF media such that the ventral part of the brain was facing towards us and the dorsal region (surface of cortex) was facing the back of the vibratome blade holder. Subsequently, acute coronal cortical sections of 20 µm thickness were prepared using the vibratome at 4.5 mm amplitude, 84–86 Hz frequency, and 2.0–2.5 mm/sec speed. Next, the membrane inserts were incubated with fresh complete growth medium containing 50% MEM (INV-42360032, Invitrogen), 25% BME (INV-21010046, Invitrogen), 5% heat-inactivated horse serum (INV-26050070, Invitrogen), 10 ng/ml nerve growth factor (NGF) (INV-A42627, Invitrogen) and glial cell line-derived neurotrophic factor (GDNF) (INV-AF-450-44, Invitrogen). The cut sections were then transferred using a sterile pasteur pipette to the inserts. For nuclear staining, slices were stained with 0.5 mM Hoechst 33342 (INV-H3570, Invitrogen) for 10 min followed by successive washing steps in ACSF media. The slices were then transferred to coverslip bottom petri dish (VWR-MSPP-P35G014C-CS, Mattek Corp MS) for their time-lapse imaging in Leica MICA fluorescent microscope (Leica Microsystems). In experiments on P2Y6 inhibition, acute brain slices were incubated for 30 min with 2 µM P2Y6 inhibitor MRS-2578 (SIG-M0319, Sigma) re-suspended in 0.1% DMSO with complete growth media, or mock alone (0.1% DMSO). For $Ca^{2+}$ imaging, slices were incubated simultaneously with 0.5 mM Hoechst and 0.5 mM Calbryte 590 AM (VWR-76484-390-EA, AAT Bioquest) for 45 min.

## Time-lapse imaging

The time-lapse imaging experiments were performed using environmental climate setup in Leica MICA for live cell imaging supplemented with 5% $CO_2$ and 37°C to ensure slice viability. Image acquisition parameters for all channels (EGFP, Hoechst, and Calbryte 590AM) were kept at 100ms exposure, with the constant focusing on the green channel (EGFP) to avoid any focal drift. All the time-lapses were subjected to a 3D z stacking of optical sections (1024*1024 pixels) with a total of 10 z-stacks at an average step size of 2 µm from the upper layer of cortex until the bottom of the 20 µm thin cortical brain slice. The entire images were captured by the 10X objective collecting the frames for every 5 min time interval for the complete 3-hr duration. Background for the videos was then deduced from Leica MICA using the thunder and lightning module. 3D representations were drawn by Leica MICA software and image processing of the time-lapse videos was further analyzed using ImageJ (FIJI).

## Microglial cell tracking and analysis

Time-lapse videos generated by Leica MICA were loaded onto ImageJ (FIJI). The number of cells included in the analysis was constrained by the number of cells available within the imaged fields of view; all cells within each field of view were included in the analysis. Using Manual tracking (*Figure 3a*) and MtrackJ plugin (*Figure 2d*), the individual microglial cells were tracked for 3 hr to obtain the quantitative differential cell mobility parameters. Briefly, length was measured by the plugin as the path length between the first and last current point of the cell track. Accumulated distance was calculated as the sum of the incremental distance measured from all the frames. Euclidean distance as the straight-line distance between the frames. Displacement as the change in position from the first to the last point of the cell track. Mean straight line speed as the total track displacement divided by the net tracking time. Velocity is measured by the ratio of accumulated distance to the total track migration time. Track mean speed as the average of all the velocities linking the various tracks. Linearity of forward progression by the ratio between the track mean straight line speed and track mean speed. Finally, the tracking coordinates for each microglial cell for various groups were represented as ross plots by importing these values into the Chemotaxis tool software (Ibidi GmbH, Germany).

## Analysis of microglial cell mobility, morphology, and calcium signal intensity using Trackmate

The datasets from time-lapse imaging in *Figure 2a, b, c*, *Figure 2—figure supplement 2*; *Figure 3b, c, d, e*, *Figures 6, 7a,b* and *Figure 8a* were analyzed using Trackmate 7.13.2 (FIJI). Microglial cell mobility, calcium signal intensity, and distance were detected by the thresholding detector, where auto-generated threshold values were used for each group. Spot filtering was carried out by the quality feature above 42.4. Each spot was linked by the advanced Kalman tracker algorithm (max frame gap = 1, alternative linking cost factor = 1.05, Kalman search radius = 20, linking max distance = 5.0, gap closing max distance = 15.0, merging max distance = 15.0, cutoff percentile = 0.9). For differentiating high- and low-mobility microglia, track displacement cutoff of 15 µm was set based on the average obtained from the mobility values of normal microglia (displacement <15 µm; low and displacement ≥15 µm; high). The mean directional change rate was calculated by averaging the successive angle values for the entire frame.

For microglial morphology analysis, we quantified morphological parameters (radius, area, perimeter, and shape index) for individual EGFP+ cells in each time frame of the time-lapse recordings using the TrackMate 7.13.2 plugin in FIJI. Parameter values for each cell were then averaged across the entire 3-hr imaging period to obtain a single mean value per cell.

## Immunofluorescence

Formalin-fixed brains (3 mm slices) were treated for 1 hr in 96% formic acid before being embedded in paraffin using standard procedures. 4 µm sections produced with Leica RM2235 microtome (Leica Biosystems, Buffalo Grove, IL) were mounted on Superfrost Plus Microscope slides (#22-037-246, Fisher Scientific, Hampton, NH) and processed for immunofluorescence according to standard protocols. To expose epitopes, slides were subjected to 20 min of hydrated autoclaving at 121°C in Citrate Buffer, pH 6.0, Antigen Retriever (#C9999, Sigma-Aldrich). Autofluorescence Eliminator Reagent (Sigma-Aldrich) and Signal Enhancer (Thermo Fisher) were used on slides according to the original protocols to reduce background fluorescence. The images were collected with Leica MICA and processed in FIJI.

## Confocal microscopy and 3D image reconstruction

Confocal images were acquired with Leica TCS SP8 microscope using laser lines 405, 488, 552, the 40×/1.30 oil immersion objective, the resolution of 1024 × 1024 pixels, and a scan speed of 400 Hz. For 3D reconstruction, the system-optimized number of steps was used. Images were processed using the LAS X and ImageJ software.

## Western blot

For Western blots, 10% (wt/vol) BH were prepared as previously described (*Makarava et al., 2012*) using RIPA Lysis Buffer (Millipore-Sigma, St. Louis, MO). To analyze brain-derived PrP$^{Sc}$, BH aliquots were diluted with RIPA buffer to achieve 5% BH final concentration and treated with 20 µg/ml proteinase K (New England BioLabs) in the presence of 50 mM Tris, pH 7.5, and 2% Sarcosyl, for 30 min at 37°C. To analyze other proteins, BH was diluted with RIPA buffer to 1% and proteinase digestion was omitted. The resulting samples were supplemented with 4xSDS loading buffer and heated for 10 min in a boiling water bath before loading onto NuPAGE 12% Bis-Tris gels. Wet transfer onto PVDF membranes and probing of Western blots was done according to standard procedures. The signals were visualized by Immobilon Forte Western HRP Substrate (Millipore-Sigma, Rockfield, MD) or SuperSignal West pico PLUS Chemiluminescent Substrate (Thermo Scientific, Rockford, IL) using Invitrogen iBright 1500 imager, and quantified with iBright Analysis software (Thermo Scientific, Rockford, IL). Intensity data were presented as normalized by actin, except for PrP$^{Sc}$ Western blots treated with protease K.

## Tracking and analysis of acutely isolated cells

Mouse microglial cells were isolated using Adult Brain Dissociation Kit with CD11b antibodies according to the manufacturer protocol (Miltenyi Biotec, #130-107-677). For non-infected B6.129P2(Cg)-*Cx-3cr1$^{tm1Litt}$*/J mice, microglia were purified from pools of three cortices, which ensured enough cells. For animals infected with SSLOW (1% BH via i.p. route), microglia were purified from individual cortices of

**Table 2.** Primer sequences for qRT-PCR.

| Primer | Accession number | Sequence |
| --- | --- | --- |
| P2ry6 | NM_183168 | F 5'-CAGTCTTTGCTGCCACAGGCAT-3' |
| | | R 5'-AGCAAGAAGCCGATGACCGTGA-3' |
| P2ry13 | NM_028808 | F 5'-TGGCATCAGGTGGTCAGTCACA-3' |
| | | R 5'-TTGTGCCTGCTGTCCTTACTCC-3' |
| P2rx7 | NM_011027 | F 5'-GAACACGGATGAGTCCTTCGTC-3' |
| | | R 5'-CAGTGCCGAAAACCAGGATGTC-3' |
| P2rx4 | NM_011026 | F 5'-GCTTTCAGGAGATGGCAGTGGA-3' |
| | | R 5'-TGTAGCCAGGAGACACGTTGTG-3' |
| Trem2 | NM_031254 | F 5'-CTACCAGTGTCAGAGTCTCCGA-3' |
| | | R 5'-CCTCGAAACTCGATGACTCCTC-3' |
| Tmem119 | NM_146162 | F 5'-ACTACCCATCCTCGTTCCCTGA-3' |
| | | R 5'-TAGCAGCCAGAATGTCAGCCTG-3' |
| Tlr2 | NM_011905 | F 5'-ACAGCAAGGTCTTCCTGGTTCC-3' |
| | | R 5'-GCTCCCTTACAGGCTGAGTTCT-3' |
| Cd68 | NM_009853 | F 5'-ACTGGTGTAGCCTAGCTGGT-3' |
| | | R 5'-CCTTGGGCTATAAGCGGTCC-3' |
| GAPDH | NM_008084 | F 5'-CATCACTGCCACCCAGAAGACTG-3' |
| | | R 5'-ATGCCAGTGAGCTTCCCGTTCAG-3' |

six mice at the clinical stage of the disease. Cells were counted and plated into wells of 24-well plates and left overnight to settle and recover from enzyme treatment. The next morning, N2a cells were added to the designated well at a 1:1 ratio, and live images were taken on the green channel at ×10 magnification for 6 hr with 5-min intervals. During the whole imaging session, cells were kept at 37°C in an atmosphere of 5% $CO_2$. For microglia motility analysis, TrackMate plugin for Fiji ImageJ software was used. All individual tracks were manually validated for the accuracy of automated tracking and corrected if needed.

## RT-qPCR

Total RNA was isolated from 10% BH in RIPA buffer. 100 µl BH aliquots were further homogenized within RNase-free 1.5 ml tubes in 200 µl of Trizol (Thermo Fisher Scientific, Waltham, MA, USA), using RNase-free disposable pestles (Fisher Scientific, Hampton, NH, USA). After homogenization, an additional 600 µl of Trizol was added to each homogenate, and the samples were centrifuged at 11,400 × $g$ for 5 min at 4°C. The supernatant was collected, incubated for 5 min at room temperature, then supplemented with 160 µl of cold chloroform and vigorously shaken for 30 s by hand. After an additional 5 min incubation at room temperature, the samples were centrifuged at 11,400 × $g$ for 15 min at 4°C. The top layer was transferred to new RNase-free tubes and mixed with an equal amount of 70% ethanol. Subsequent steps were performed using an Aurum Total RNA Mini Kit (Bio-Rad, Hercules, CA, USA) following the manufacturer instructions. Isolated total RNA was subjected to DNase I digestion. RNA purity and concentrations were estimated using a NanoDrop One Spectrophotometer (Thermo Fisher Scientific, Waltham, MA). Complementary DNA (cDNA) synthesis was performed using iScript cDNA Synthesis Kit as described elsewhere. The cDNA was amplified with CFX96 Touch Real-Time PCR Detection System (Bio-Rad, Hercules, CA) using SsoAdvanced Universal SYBR Green Supermix and the primers listed in *Table 2*. The PCR protocol consisted of 95°C incubation for 2 min followed by 40 amplification cycles at 95°C for 5 s and 60°C for 30 s. The data were analyzed using CFX96 Touch Real-Time PCR Detection System Software.

## Study approval

The study was carried out in strict accordance with the recommendations in the Guide for the Care and Use of Laboratory Animals of the National Institutes of Health. The animal protocol was approved by the Institutional Animal Care and Use Committee of the University of Maryland, Baltimore (Assurance Number: D16-00125; Protocol Number: AUP-00000166).

## Statistical analysis

All statistical analysis and graph plotting were performed using GraphPad Prism software, version 10.1.1 for Windows.

For data presented in *Figures 2a, b, 3c-e, 6a-d, 7a*, *Figures 8a, b and 9a-c* and *Figure 2—figure supplement 2*, mean values and statistical significance were calculated at the single-cell level ($n$ = number of cells analyzed; $N$ = number of animals). For data presented in *Figure 3b*, *Figure 1—figure supplement 1e*, *Figure 2—figure supplement 1b*, *Figure 8—figure supplement 1b*, mean values and statistical significance were calculated at the level of biological replicates (animals), where $N$ = number of animals analyzed. For the Superplots (*Figure 3—figure supplement 1*), mean values and statistical significance were likewise calculated based on biological replicates (animals), with $N$ representing the number of animals.

## Acknowledgements

We thank Kara Molesworth for helping with animal procedures, and Olga Mychko for RNA isolation and immunofluorescence. Financial support for this study was provided by National Institutes of Health Grants R01 NS045585 and R01 NS129502 to IVB.

## Additional information

### Funding

| Funder | Grant reference number | Author |
| --- | --- | --- |
| National Institutes of Health | NS045585 | Ilia V Baskakov |
| National Institutes of Health | NS129502 | Ilia V Baskakov |

The funders had no role in study design, data collection, and interpretation, or the decision to submit the work for publication.

### Author contributions

Sunitha Subhramanian, Conceptualization, Data curation, Formal analysis, Validation, Investigation, Visualization, Methodology, Writing – review and editing; Olga Bocharova, Formal analysis, Validation, Investigation, Visualization, Methodology, Writing – review and editing; Natallia Makarava, Formal analysis, Supervision, Investigation, Writing – review and editing; Tarek Safadi, Formal analysis, Investigation, Writing – review and editing; Ilia V Baskakov, Conceptualization, Data curation, Supervision, Visualization, Writing – original draft, Project administration, Writing – review and editing

### Author ORCIDs

Ilia V Baskakov ⬤ https://orcid.org/0000-0003-2821-0942

### Ethics

The study was carried out in strict accordance with the recommendations in the Guide for the Care and Use of Laboratory Animals of the National Institutes of Health. The animal protocol was approved by the Institutional Animal Care and Use Committee of the University of Maryland, Baltimore (Assurance Number: D16-00125; Protocol Number: AUP-00000166).

Reviewer #1 (Public review): https://doi.org/10.7554/eLife.107650.3.sa1
Reviewer #2 (Public review): https://doi.org/10.7554/eLife.107650.3.sa2

Author response https://doi.org/10.7554/eLife.107650.3.sa3

## Additional files

### Supplementary files
MDAR checklist

Source data 1. Source data for the Figures containing graphs/plots.

### Data availability
All data supporting the findings of this study are available within the paper and its Supplementary Information.

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
