## [Editor Report · eLife Assessment]

This **fundamental** study provides new evidence of a change in how microglia survey neurons during the chronic phase of neurodegeneration, which researchers studying neuroinflammation and its role in neurodegenerative disease should find interesting. In this research, using time-lapse imaging of acute brain slices from prion-affected mice, the researchers show that, unlike in healthy brains, microglia become reactive, lose their territorial boundaries, and become highly mobile, exhibiting "kiss-and-ride" behavior, migrating into brain tissue and forming reversible, transient body-to-body contact with neurons. The evidence is **compelling**, with well-executed time-lapse imaging, good quantitative analysis across several disease stages, pharmacological validation of P2Y6 involvement, and the very surprising finding that this mobile behavior persists after microglia are removed from the brain.

---

## [Referee Report · Reviewer #1 (Public review)]

Summary:

In this manuscript, Subhramanian et al. carefully examined how microglia adapt their surveillance strategies during chronic neurodegeneration, specifically in prion-infected mice. The authors used ex vivo time-lapse imaging and in vitro strategies and found that reactive microglia adopt a highly mobile, "kiss-and-ride" behavior, contrasting the more static surveillance typical of homeostatic microglia. The manuscript provides fundamental mechanistic insights into the dynamics of microglia-neuron interactions, implicates P2Y6 signaling in regulating mobility, and suggests that intrinsic reprogramming of microglia might underlie this behavior, the conclusions are therefore compelling.

Strengths:

(1) The novelty of the study is high, particularly the demonstration that microglia lose territorial confinement and dynamically migrate from neuron to neuron under chronic neurodegeneration.

(2) The possible implications of a stimulus-independent high mobility in reactive microglia are particularly striking. Although this is not fully explored.

(3) The use of time-lapse imaging in organotypic slices rather than overexpression models provided a more physiological approach.

(4) Microglia-neuron interactions in neurodegeneration have broad implications for understanding the progression of diseases, such as Alzheimer's and Parkinson's, that are associated with chronic inflammation.

Weaknesses:

Previous weaknesses were addressed.

---

## [Referee Report · Reviewer #2 (Public review)]

This is a nice paper focused microglial responses to different clinical stages of prion infection in acute brain slices. The key here is the use of time-lapse imaging that captures the dynamics of microglial surveillance, including morphology, migration, and intracellular neuron/microglial contacts. The authors use a myeloid GFP-labeled transgenic mouse to track microglia in SSLOW-infected brain slices, quantifying differences in motility and microglial-neuronal interactions via live fluorescence imaging. Interesting findings include the elaborate patterns of motility among microglia, the distinct types and durations of intracellular contacts, the potential role of calcium signaling in facilitating hypermobility, and the fact that this motion-promoting status is intrinsic to the microglia, persisting even after the cells have been isolated from infected brains. Although largely a descriptive paper, it offers mechanistic insights, including the role of calcium in supporting microglial movement, with bursts of signaling identified even within the time lapse format, and inhibition studies implicating the purinergic receptor and calcium transient regulator P2Y6 in migratory capacity.

Strengths:

(1) The focus on microglia activation and activity in the context of prion disease is interesting

(2) Two different prions produce largely the same response

(3) Use of time-lapse provides insight into the dynamics of microglia, distinguishing between types of contact - mobility vs motility - and providing insight on the duration/transience and reversibility of extensive somatic contacts that include brief and focused connections in addition to soma envelopment.

(4) Imaging window selection (3 hours) guided by prior publications documenting preserved morphology, activity, and gene expression regulation up to 4 hours.

(5) The distinction between high- and low-mobility microglia is interesting, especially given that hypermobility seems to be an innate property of the cells.

(6) The live-imaging approach is validated by fixed tissue confocal imaging.

(7) The variance in duration of neuron/microglia contacts is interesting, although there is no insight into what might dictate which status of interaction predominates

(8) The reversibility of the enveloping action, which is not apparently a commitment to engulfment, is interesting, as is the fact that only neurons are selected for this activity.

(9) The calcium studies use the fluorescent dye calbryte-590, which picks up neuronal and microglial bursts -prolonged bursts are detected in enveloped neurons and in the hyper-mobile microglia - the microglial lead is followed up using MRS-2578 P2Y6 inhibitor that blunts the mobility of the microglia

Comments on revisions:

The authors have addressed my concerns in full - I think this is a very nice addition to the literature.

---

## [Author Response]

The following is the authors’ response to the original reviews.

**Public Reviews:**

**Reviewer #1 (Public review)**
The Cx3cr1/EGFP line labels all myeloid cells, which makes it difficult to conclude that all observed behaviors are attributable to microglia rather than infiltrating macrophages. The authors refer to this and include it as a limitation. Nonetheless, complementary confirmation by additional microglia markers would strengthen their claims.

We appreciate the reviewer’s insightful comment regarding the cellular identity of the enveloping myeloid cells. As suggested, we performed triple co-immunostaining of SSLOW-infected Cx3cr1/EGFP mice using markers for neurons (NeuN), myeloid cells (IBA1), and resident microglia (TMEM119 or P2Y12). Because formic acid treatment used to deactivate prions abolishes the EGFP signal, we relied on IBA1 staining to identify the myeloid population. Our results confirmed that IBA1⁺ cells exhibiting the envelopment behavior are also TMEM119⁺ and P2Y12⁺, consistent with a resident microglial phenotype. These new data are presented in Figures S3 and S4 and described in the final section of the Results.

Although the authors elegantly describe dynamic surveillance and envelopment hypothesis, it is unclear what the role of this phenotype is for disease progression, i.e., functional consequences. For example, are the neurons that undergo sustained envelopment more likely to degenerate?

We appreciate this important question regarding the functional implications of neuronal envelopment. At present, technical limitations prevent us from continuously tracking the fate of individual enveloped neurons in prion-infected mice. Nevertheless, our recent study demonstrated that P2Y12 knockout increases the prevalence of neuronal envelopment and accelerates disease progression (Makarava et al., 2025, J. Neuroinflammation). These findings suggest that while microglial envelopment may represent an adaptive response to increased neuronal surveillance demands, excessive envelopment, as observed in the absence of P2Y12, appears to be maladaptive. A new paragraph has been added to the Discussion to address this point.

Moreover, although the increase in mobility is a relevant finding, it would be interesting for the authors to further comment on what the molecular trigger(s) is/are that might promote this increase. These adaptations, which are at least long-lasting, confer apparent mobility in the absence of external stimuli.

We thank the reviewer for this thoughtful suggestion. The molecular mechanisms underlying the increased mobility of microglia in prion-infected brains remain to be identified, and we plan to pursue this question in future studies. One possibility we briefly discuss in the revised manuscript is that proinflammatory signaling, mediated by secreted cytokines or interleukins, may drive this phenotype. Supporting this hypothesis, recent work has shown that IFNγ enhances microglial migration in the adult mouse cortex (doi:10.1073/pnas.2302892120). This work has been cited in the revised manuscript.

The authors performed, as far as I could understand, the experiments in cortical brain regions. There is no clear rationale for this in the manuscript, nor is it clear whether the mobility is specific to a particular brain region. This is particularly important, as microglia reactivity varies greatly depending on the brain region.

We appreciate this insightful comment highlighting the importance of regional determinants of microglial reactivity, which indeed aligns with our ongoing research interests. In our previous studies, neuronal envelopment by microglia was observed consistently across all prion-affected brain regions exhibiting neuroinflammation. Assuming that envelopment requires microglial mobility, it is reasonable to speculate that microglia are mobile in all brain regions affected by prions and displaying neuroinflammatory responses. In the current study, we focused exclusively on the cortex because this region was used for quantifying the prevalence of neuronal envelopment as a function of disease progression in our prior work (DOI: 10.1172/JCI181169), which guided the present study design. Our ongoing investigations indicate that the prevalence of envelopment is region-dependent and correlates with microglial reactivity/the degree of neuroinflammation. In prion diseases, the degree of microglial reactivity is dictated by the tropism of specific prion strains to distinct brain regions. Notably, our prior studies have shown that strain-specific sialylation patterns of PrP^Sc^ glycans play a key role in determining both regional strain tropism and the extent of neuroinflammatory activation (DOI: 10.3390/ijms21030828, DOI: 10.1172/JCI138677). In response to this comment, we have added a brief rationale for using the cortex in the Results section.

It would be relevant information to have an analysis of the percentage of cells in normal, sub-clinical, early clinical, and advanced stages that became mobile. Without this information, the speed/distance alone can have different interpretations.

We thank the reviewer for this valuable suggestion. The percentage of mobile cells across normal, sub-clinical, early clinical, and advanced disease stages is presented in Figure 3b and described in the final paragraph of the section “Enveloping behavior of reactive myeloid cells.”

**Reviewer #2 (Public review)**
The number of individual cells tracked has been provided, but not the number of individual mice. The sex of the mice is not provided.

We used N = 3 animals per group throughout the study; this information has now been added to the figure legends. Animals of both sexes were included in random proportions. The sex information is now listed for each experiment in the Animals subsection of the Methods.

The statistical approach is not clear; was each cell treated as a single observation?

Yes, with the exception of the heat map in Figure 2d, all mobility parameters are analyzed and presented at the level of individual cells, with each cell treated as an independent observation. The primary aim of this study is to characterize behavioral patterns of single reactive myeloid cells. Analyzing data at the cell level allows us to capture the full distribution of cell behaviors and to preserve biologically meaningful heterogeneity within and across animals. By contrast, averaging values per animal would largely mask this variability. In the heat map in Figure 2d, data are averaged per animal, specifically to illustrate inter-animal variability within each group and to visualize changes across disease progression**.**

The potential for heterogeneity among animals has not been addressed.

To address this concern, we now include a new Supplemental Figure (Figure S4) presenting the data using Superplots, in which individual cells are shown as dots, animal-level average as circles, and group means calculated based on animals as black horizontal lines. These plots demonstrate that cell mobility measures are highly consistent across animals within each group, indicating limited inter-animal heterogeneity.

Validation of prion accumulation at each clinical stage of the disease is not provided.

We now provide validation of PrP^Sc^ accumulation across disease stages by Western blot, along with quantitative analysis, in a new Supplemental Figure (Figure S2). This confirms progressive PrP^Sc^ accumulation with advancing disease.

How were the numerous captures of cells handled to derive morphological quantitative values? Based on the videos, there is a lot of movement and shape-shifting.

The following description has been added to Methods to clarify morphology analysis: For microglial morphology analysis, we quantified morphological parameters (radius, area, perimeter, and shape index) for individual EGFP⁺ cells in each time frame of the time-lapse recordings using the TrackMate 7.13.2 plugin in FIJI. Parameter values for each cell were then averaged across the entire three-hour imaging period to obtain a single mean value per cell.

While it is recognized that there are limits to what can be measured simultaneously with live imaging, the authors appear to have fixed tissues from each time point too - it would be very interesting to know if the extent or prion accumulation influences the microglial surveillance - i.e., do the enveloped ones have greater pathology.

This is very interesting question which is difficult to answer due to technical challenges in monitoring the pathology or faith of individual neuronal cells as a function of their envelopment in live prion-infected animals. Our previous work revealed that both accumulation of total PrP^Sc^ in a brain and accumulation of PrP^Sc^ specifically in lysosomal compartments of microglia due to phagocytosis precedes the onset of neuronal envelopment (DOI: 10.1172/JCI181169). Moreover, the onset of neuronal envelopment occurred after a noticeable decline in neuronal levels of Grin1, a subunit of the NMDA receptor essential for synaptic plasticity. Reactive microglia were observed to envelop Grin1-deficient neurons, suggesting that microglia respond to neuronal dysfunction. However, considering that envelopment is very dynamic and - in most cases - reversible, correlating the degree of envelopment with dysfunction of individual neurons is technically challenging.

**Recommendations for the authors**

**Reviewer #1 (Recommendations for the authors):**
(1) I recommend performing additional immunostaining using microglial markers to address specificity.

These new data showing immunostaining for markers of resident microglia TMEM119 and P2Y12 are presented in Figures S6 and S7 and described in the final section of the Results.

(2) The authors can at least further discuss the functional consequences of their findings in further detail.

A new paragraph has been added to the Discussion to address this point.

(3) Quantify the % of cells that become mobile in the different conditions.

The percentage of mobile cells across normal, sub-clinical, early clinical, and advanced disease stages is presented in Figure 3b and described in the final paragraph of the section “Enveloping behavior of reactive myeloid cells.”

(4) Improve method details on the brain regions used and further expand the statistical section.

We have expanded the Statistical Analysis section to indicate whether statistical comparisons and mean values were calculated at the single-cell level or the animal level for each analysis. The specific statistical tests used and the number of animals (N) are now reported in the corresponding figure legends. The sex of animals is provided in Table 1 (Methods). Only the cortical region was examined in this study; this information is stated in the Methods and is now also noted in the figure legends for clarity.

**Reviewer #2 (Recommendations for the authors):**
(1) More details on members of the PY2 receptor family expressed in microglia would be helpful. The study highlights a previously published prion-induced decline in the expression of P2Y12, a microglial marker that is required for intracellular neuron-microglial contacts, and P2Y6, involved in calcium transients, which is required for hypermotility. How are members of this family of receptors regulated at the gene and/or protein level in microglial and given their responsiveness to nucleotide ligands, are other members implicated in the properties being quantified here?

We appreciate the reviewer’s insightful comment. To address this point, we examined the expression of multiple P2Y receptors and ATP-gated P2X channels known to contribute to microglial surveillance, activation, motility, and phagocytosis, alongside the activation markers Tlr2, Cd68, and Trem2. Bulk brain transcript analyses indicated that all examined genes were upregulated in SSLOW-infected mice relative to controls (new Figure S5a). However, because microglial proliferation substantially increases microglial numbers during prion disease progression, bulk tissue measurements do not necessarily reflect per-cell expression levels. Therefore, we normalized gene expression values to the microglia-specific marker Tmem119, whose per-cell expression remains stable across disease stages (Makarava et al., 2025, J. Neuroinflammation). After normalization, Tlr2, Cd68, and Trem2 were increased approximately 10-, 6-, and 4-fold, respectively. In contrast, P2 receptor genes showed more modest changes: P2ry6 increased ~3-fold, P2ry13 ~2-fold, and P2rx7 ~1.3-fold, while P2rx4 remained unchanged (Figure S5a). Within the scope of the present study, we focused on P2Y6 due to (i) its role in regulating calcium transients, (ii) the magnitude of its upregulation relative to other P2 receptors, and (iii) its highly microglia-specific expression in the CNS. We note that currently available commercial P2Y6 antibodies lack sufficient specificity, making reliable assessment of protein-level expression challenging.

(2) Is P2Y6 expressed in any other cell type that might account for the blunted mobility of the microglia? The authors mention P2Y12 also identifies the GFP cells; however, it would be beneficial to highlight the specificity of the target in the ex vivo treatment of the infected slices.

In the brain, both P2Y12 and P2Y6 are considered highly specific to resident microglia under physiological and neuroinflammatory conditions. P2Y12 is, in fact, widely used as a canonical marker of homeostatic and resident microglia. While P2Y6 is also expressed in peripheral myeloid cells such as macrophages, our phenotypic characterization indicates that the cells exhibiting neuronal envelopment are TMEM119⁺ and P2Y12⁺, consistent with a resident microglial identity. These data, including new analyses added to the revised manuscript, support that the cells responding to P2Y6 signaling in our ex vivo slice experiments are resident microglia.

(3) The fluorescent mouse lacks Cx3cr1 - have the authors investigated why there were no apparent consequences, at least in the context of prion infection? Are there functional redundancies that might be harnessed? Does this impact the generalizability of the findings here?

The role of Cx3cr1 in prion disease has been directly examined in two independent studies (doi: 10.1099/jgv.0.000442; doi: 10.1186/1471-2202-15-44). One study reported no effect of Cx3cr1 deficiency on disease incubation time, whereas the other observed only a minor difference. Importantly, both studies found no detectable alterations in microglial activation patterns, cytokine expression, or PrP^Sc^ deposition in Cx3cr1-deficient mice compared to wild-type controls. Our own data (Figure S1) are consistent with these findings: disease course and PrP^Sc^ deposition were comparable between Cx3cr1/EGFP and wild-type mice. Moreover, we observed reactive microglial envelopment of neurons in both genotypes. Microglia isolated from SSLOW-infected Cx3cr1/EGFP mice also displayed similarly elevated mobility in vitro, in agreement with our previous observations of high mobility of microglia isolated from SSLOW-infected wild-type mice (Makarava et al., 2025, J. Neuroinflammation). Taken together, these results indicate that Cx3cr1 is not a key determinant of reactive microglial mobility or envelopment behavior in prion disease. Thus, the use of the Cx3cr1/EGFP reporter line does not compromise the generalizability of our conclusions.

(4) The distinction between high mobility and low mobility microglia is interesting - is there any evidence to suggest that the slow-moving microglia are actually a separate class - do enveloping microglia exhibit both mobility states - can the authors comment on plasticity here?

We appreciate this insightful comment, which closely aligns with our ongoing interests. At present, we do not have evidence to support that high- versus low-mobility microglia represent distinct molecular phenotypes. Given that our time-lapse imaging spans only a three-hour window, it remains unclear whether these mobility states reflect stable cell-intrinsic properties or transient phases within a dynamic surveillance process. Notably, we observed that individual cells can transition between more stationary, neuron-associated states and highly mobile states within the same imaging session. In future work, we intend to investigate whether prolonged interactions with neuronal somas or other microenvironmental cues may drive diversification of reactive myeloid cell phenotypes.

(5) In the discussion, the authors speculate about "collective coordinated decision making" - that seems a stretch unless greater context is provided. The fact that several microglia can be found in contact with an individual neuron and that each microglia can connect with multiple neurons simultaneously is certainly interesting; however, evidence for hive behavior is entirely lacking.

We agree with the reviewer that our previous wording overstated the interpretation. The statement regarding collective decision-making has been removed.